# A paradigm shift in learning strategy research: Illustration and example of a within-person examination

**Tsuyoshi Yamaguchi***

Liberal Arts and Sciences, Nippon Institute of Technology, Saitama Pref., Japan

* yamaguchi.tsuyoshi@nit.ac.jp

## Abstract

Although research on learning strategies has contributed to improving learners' performance, most studies only described the characteristics of learners. These approaches have focused on inter-individual differences and their correlations, making it difficult to examine the process of selecting learning strategies in individuals. To examine the correlations within individual learners, such as the variables that influence the use of a strategy by a specific learner, two surveys were conducted in this study. Perceived cost and perceived benefit of using each strategy as the influential variables on choosing a strategy were adopted with reference to decision-making research. Perceived cost consisted of the time perspective for learning outcomes by the learning strategy and the method of learning strategy implementation. In Study 1, cognitive strategies that have a direct impact on the acquisition of learning content were measured, and metacognitive strategies that monitor and regulate one's own learning behavior were examined in Study 2. The variables were treated for each individual to calculate intra-class and within-person correlations, although data were collected using a typical questionnaire survey in learning strategy studies. Hierarchical Bayesian modeling was conducted with use of the learning strategy as the objective variable and the perceived costs and benefits as explanatory variables. The analysis revealed that, in terms of cognitive strategies (Study 1), learners avoided using learning strategies perceived to have a high cost in the short term; however, they used cognitive strategies regardless of the circumstances. Furthermore, regarding metacognitive strategies (Study 2), students avoided using learning strategies that they perceived as costly to use in the short term and used them only when they suited the circumstances. On the other hand, in metacognitive strategies (Study 2), students avoided using learning strategies that they perceived as costly to use for the next test and use as appropriate for the situation. Focusing on within-individual variance and correlation made the interpretation of factors that influence the choice of learning strategies more intuitive and provided more suggestions for educational practice.

**Data Availability Statement:** All data and analysis files are available from the Open Science Framework (OSF) project (https://osf.io/kunpg/?view_only=845e8a2b81b04db686c319c81e003420).

**Funding:** This work was supported by JSPS KAKENHI (Grant Number 13J04514 & 21K13695) and Special Grant by Nippon Institute of Technology (2019). The funders had no role in study design, data collection and analysis, decision to publish, or preparation of the manuscript.

**Competing interests:** The authors have declared that no competing interests exist.

## Introduction

The study of learning strategies has made many contributions to educational practice. It is known that there are various types of learning strategies and that elaboration and metacognitive strategies can positively predict academic performance [1]. Specifically, learners who attempt to process information deeply enough to understand what they are learning, and those who try to control their learning behavior by understanding their current learning situation, perform better. Moreover, learning strategies have become a key component of research frameworks for shaping self-regulated learning (SRL): Recent reviews have proposed several theoretical models for SRL, all of which assume that learning strategies play some role [2], and some theories explicitly incorporate specific learning strategies into their models [3–5]. However, most related research has examined inter-individual differences, yielding results and interpretations such as "A learner j with higher levels of learning characteristic x is more likely to use y in strategy k (e.g., Learners with a high perceived cost of a learning strategy [I focus on memorizing the facts without considering why] tend not to use that strategy as much as learners with a low cost perception [6])" (see Fig 1). This perspective is especially important in describing the characteristics of self-regulated learners. However, an equivalent but more specific intra-individual perspective, such as "The use of y in strategy k for given learner j

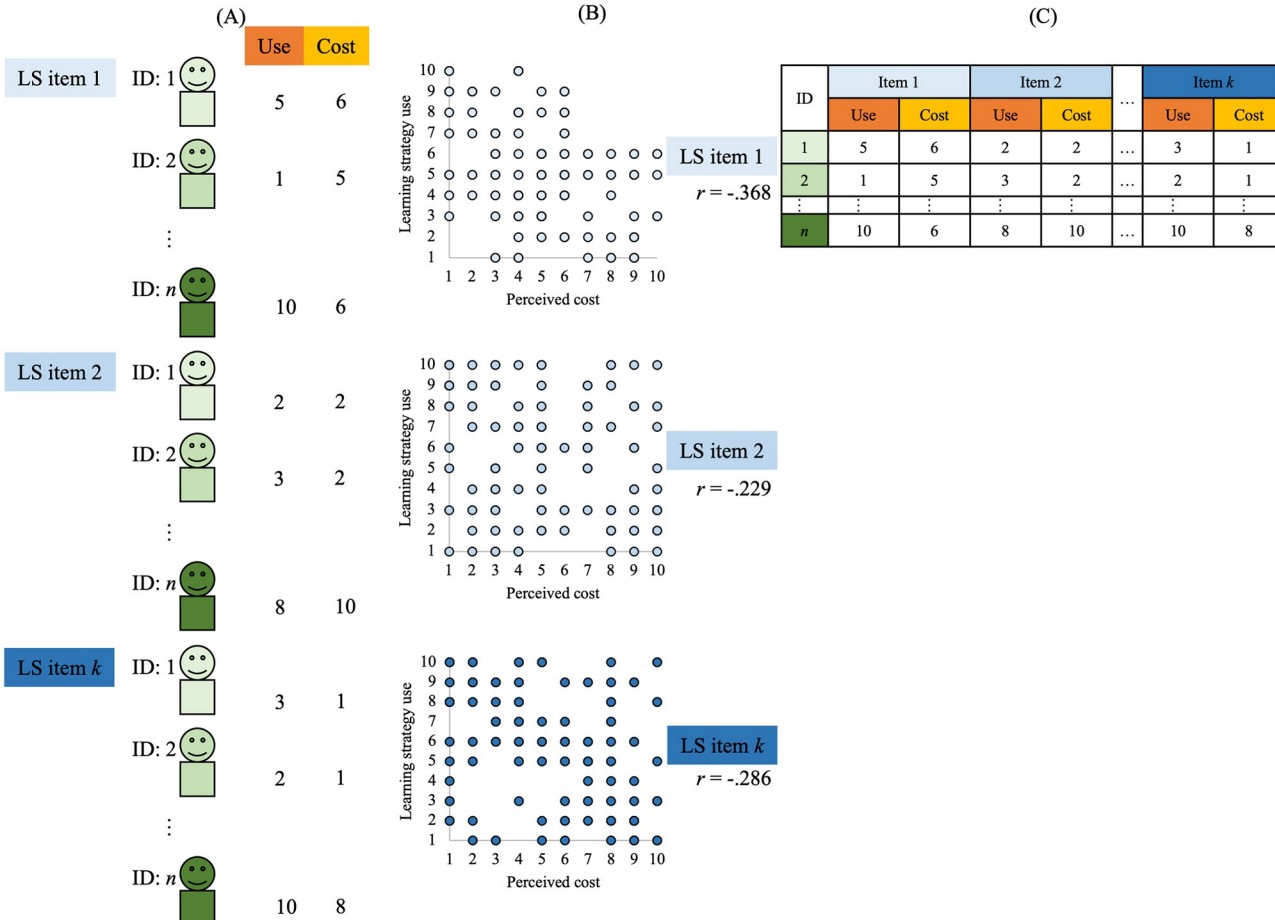

**Fig 1. Image, dataset, and example analysis of data acquisition from a study dealing with variance and covariance between individuals.** (A) For each item of a strategy, scores for each participant's use and cost perception of that strategy are collected, and (B) a scatterplot is drawn and correlation coefficients are calculated. (C) An image of the actual data set.

increases when intra-personal variable x is higher (e.g., If the perceived cost of learning strategy [When I learn a new historical event, I try to understand what actually happened] is higher than the perceived cost of learning strategy [I focus on memorizing the facts without considering why] within a single learner, the more costly learning strategy will not be used [6]),” may be essential to promote the use of learning strategies (see Fig 2). As self-regulated learners naturally employ multiple learning strategies [3], their flexibility in selecting and implementing strategies appropriate to the content and task is revealed by focusing on the individual learner's use of varied strategies. Thus, the purpose of this study was to provide a within-person perspective on the use of learning strategies, such as why a learner used a specific learning strategy out of multiple options. Therefore, the present research did not focus on the effects of specific strategies or individual differences among learners, and was not concerned with the type of strategies used, as it is desirable that there be variance in the strategies used within individuals. In this study, two surveys were conducted, each measuring cognitive and metacognitive strategies. The cognitive strategies focused on the levels of memory processing and includes a deep processing strategy focusing on understanding the meaning of information in the content to be learned, and a shallow processing strategy focusing on memorizing the text of the content [7]. The metacognitive strategies consisted of monitoring, in which the learner objectively grasps and evaluates their learning behavior, and control, in which the learner changes and maintains their learning behavior and sets behavioral guidelines [8]. Although the strategies

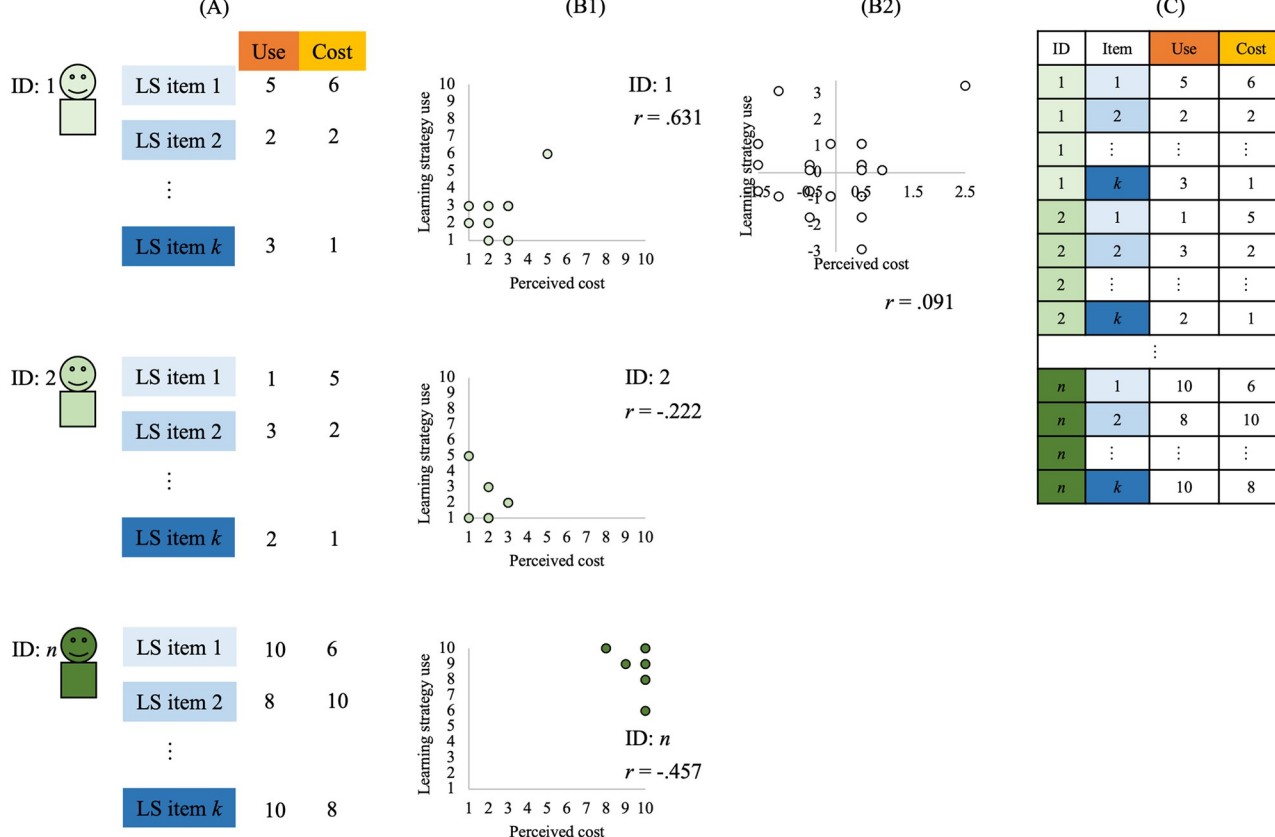

**Fig 2. Image of data acquisition for a study dealing with intra-individual variance and covariance, and an example data set and analysis.** (A) The use and cost-sensitivity scores of each strategy item are collected for each participant; (B1) a scatterplot is drawn for each participant and correlation coefficients are also calculated. (B2) The differences in the mean values of the variables are taken for each participant and combined into a single scatter plot and correlation coefficients. (C) An image of the actual dataset.

may vary from those measured in empirical studies on SRL [9–12], this study is one of the few that provides a perspective on and way to examine the within-person (between-strategy) variance and covariance of learning strategies; the items were measured based on this.

In focusing on within-person variance in the use of learning strategies, we need to be clear about the factors that explain that variance. Here, we will refer individually to the traditional aspects of metacognitive knowledge [13] in the framework of memory and decision making (the idea is to add findings from memory and decision-making research to the stages of instruction proposed in the Japanese literature on psychology for college students who want to become teachers [14]). First, the learner needs to know the strategy itself: This is called knowledge regarding strategy. When a learner uses a strategy consciously, it is thought that he or she is using it from semantic memory in long-term memory (episodic memory in the initial state). Second, we need to not only know the learning strategies but also use them. This is called procedural knowledge, involves using the learning strategy and procedural memory, which are related to the acquisition of physical and cognitive skills in long-term memory. Although there are easy learning strategies that can be used simply by knowing them, repeated experience is considered necessary to use them automatically while maintaining the characteristics of procedural memory [15]. Third, even if it is possible to use a learning strategy, the actual use may depend on what the learner subjectively thinks of the strategies. The human decision-making process is known to take into account post-action costs and benefits [16]. In the framework of learning strategies, post-action costs and benefits are not as defined as in economic behavior, and there is an element of subjectivity that depends on the learner. In fact, it has been reported that what may be effective for the researcher may not be perceived as such by the learner [9, 10]. The subjective costs and benefits that determine the use of such learning strategies in this study will be described as the perceived costs and perceived benefits. Finally, the subjective costs and benefits of the same learning strategy may vary depending on how it is used and when the learning outcomes are obtained. The usage and timing of such learning strategies are collectively referred to as conditional knowledge. Even if there is no scientific evidence, learners may change the extent to which they use a learning strategy based on their perception of its unique costs under certain conditions. (It is known that even in economic decision making, where the balance of payments after a choice is clear, the conditions under which the information is presented can make a human decision less rational [17]).

Learning strategies are cognitive activities that are invisible to the eye, so thinking in terms of screwdrivers may make it easier to understand the stages of learning strategy use. A person who sees a screwdriver for the first time may not understand its purpose (step of knowledge about tool), and even if they understand that it is a tool for tightening and loosening screws, they may be unable to use it immediately (step of procedural knowledge). Even if the screw is a Phillips-head, if a Phillips screwdriver cannot be used, a flathead screwdriver may be used instead (step of cost-benefit knowledge), especially if a Phillips screwdriver is not readily available (step of conditional knowledge about cost-benefit). Moreover, there are various shapes of screws, and there is a suitable screwdriver for each of them, allowing individuals to use them on their own. In the same way, there are a variety of learning materials and learning strategies suitable for materials, and learners are expected to use the strategies in accordance with the above mentioned steps. Therefore, we argue that it is necessary to pay attention to the within-person (between-strategy) variance in the use of learning strategies.

Thus, in the framework of learning strategies research, this study refers to the importance of within-learner perspectives, such as the fact that individuals can use multiple strategies and how they choose which strategies to use. On the other hand, it is also true that learning strategies are a major component of SRL, which is the mainstream in the fields of cognitive and educational psychology. Therefore, we will consider how the use of learning strategies within an

individual relates to the framework of SRL. According to Panadero's review [2], the concept of learners using multiple learning strategies seems to be consistent with Boekaerts' model in that it assumes multiple cognitive strategies (for learning content) [3, 4], and with the traditional Zimmerman model in that it assumes a variety of metacognitive activities (which is consistent with most SRL models) [5]. In addition, the subjective cost/benefit judgments of learning strategies that are characteristic of this study have already been reported in several SRL studies: It should be noted that this is subjective, as some studies have shown that not only learners but also instructors evaluate the benefit of strategies that are not supported by empirical studies [9–11]. It has been shown that such subjective perceptions of benefit do not necessarily change when correct knowledge is taught, that is, perceived benefit promotes the use of the learning strategy, but false efficacy of assumptions may lead to inadequate performance (on the other hand, it has also been shown that careful teaching by analyzing the details of learners' assumptions can change them to suitable perceptions of efficacy) [18]. Similarly, it has been reported that the effect may be due to a perceived cost rather than a perceived benefit. It has been reported that even if students think it is effective to study much earlier before the exam (i.e., high perceived benefit but also high perceived cost), they end up starting to study about a day before the exam (low perceived benefit and cost) [19]. There have also been reports on SRL failures, such as abandoning opportunities for restudying and retrieval practice, even though they are insufficient for later recall [20]. This suggests that, not unlike general decision-making behavior, learners may prioritize the low cost of learning strategies [21]. Thus, the variance among strategies within an individual learner and the four steps to the use of learning strategies that this study focuses on are consistent with the SRL model and its empirical studies.

## Methods background

Unsurprisingly, some social science research, including some longitudinal learning studies, have reported the importance of focusing on intra-individual changes over time [22]. Obergriesser and Stoeger, for example, measured the effectiveness of a cognitive learning strategy, interest, and boredom in eight graders using 25 texts [23]. The text measured per individual was designated Level 1 (i.e., lying on one line of the data set), those individuals who nested each measurement were designated Level 2, and the class to which each individual belonged was designated Level 3. They found that the students effectively used cognitive strategies for texts of high interest. Additionally, Glogger and colleagues measured cognitive and metacognitive strategies for nine-graders over a 6-week period [24]. They focused on the relationship between cognitive strategies and academic performance but did not examine within-individual variance because they set individual differences to hierarchal level 1: If we do not assume hierarchy in our data, as it is traditionally done, and consider the correlation between two variables with one row of the dataset as a single participant, then we are considering the variance between individuals. Similarly, if a dataset contains one participant's data in one row, even if there is hierarchy (higher levels), we cannot examine the variance within individuals, because even the smallest unit can only represent the individual differences. Hence, research on learning strategies has begun to address variation and variance within individuals while recognizing the importance of exploring intra-individual strategies. In particular, the contribution of these studies is to reveal the change in the same strategy over time and the variance in each content of the same strategy. On the other hand, it would be difficult to give any indication of the variance that can occur within an individual between multiple strategies at the same time and in the same situation, that is, within-person perspectives of learning strategy use.

One potential barrier to studying intra-personal learning strategy use is the question of how we observe and measure the flexibility of use within a person. Although Glogger and colleagues

did indeed measure multiple strategies, they performed a cluster analysis using the degree of use of those strategies to identify distinctive clusters of learners who used each strategy with similar frequency [24]; this approach only reveals group characteristics and does not examine the strategy variance of particular individuals. Specifically, by asking participants to describe their use of multiple learning strategies, the tendency to use multiple strategies within an individual is revealed. This condition is necessary for measuring within-individual variance; however, it is not unique in learning strategy research. In the datasets and examples of correlations shown in Figs 1 and 2, multiple items of the same learning strategy are measured in each case (all in Panel B). What differs is the resulting dataset and its analysis (and, of course, the accompanying considerations). In the traditional method, each row describes one person's data (Fig 1C). Alternatively, to focus on each strategy of within-individual variance of strategies, each row describes the strategy items (Fig 2C). Furthermore, the analyses require different interpretations, such as inter-individual correlation (Fig 1B) and intra-individual correlation (Fig 2B). Murayama and colleagues argued for the need for an intra-individual approach in educational psychology: As an example, they mentioned inter- and intra-personal approaches to the use of learning strategies [6]. Their paper discussed data from a questionnaire survey measuring the perceived benefit, perceived cost, and usage of a learning strategy on 17 different learning strategy items using a Likert scale [25]. Conditional knowledge, such as "when is the learning directed at," was taken into account in the perceived benefit, and the inter- and intra-individual approaches were described with the use of the learning strategy as the objective variable and the subjective perceptions of the three learning strategies as explanatory variables, such as "is the strategy effective for learning for the next test (short-term utility)," "is the strategy effective for future learning (long-term utility)," and "is the strategy cumbersome to use (cost)." Depending on the research question, such as "promoting the use of learning strategies," we can address the question more directly, such as what strategies are used most often, with a within-person approach to "compare across strategies," rather than with a between-person approach to "compare across learners" to learn who uses a particular strategy most frequently. Yamaguchi used a questionnaire survey to measure 15 items (8 cognitive strategies + 7 meta-cognitive strategies) of various learning strategies for obtaining an interpretation of the within-person approach presented by Murayama and colleagues, and proposed a 2 × 2 perceived benefit by adding conditional knowledge of usage inspired by the SRL framework, such as "how to use it effectively (use it without considering the learning content OR use it according to the learning content)," to the temporal conditional knowledge of "when is it effective (short term [next test] OR long term [future])" that they showed [26]. Thus, by focusing on within-individual variance-correlation, it is possible to clarify an individual's internal process around learning strategy use, such as why the learner uses a particular strategy.

In examining the factors that govern the use of learning strategies, as discussed above, perceived benefit has been addressed, and some studies report that learners do not to use theoretically effective strategies [9–12, 18–20]. People have been shown to value loss aversion more than profit-making, in general [27]. For instance, even if a subject is in a situation with multiple possible action options, if the outcomes are the same, they will choose the method that requires the least amount of effort [21, 28]. Garner reported that when simple no-cost strategies can produce results, they are preferred [29]. However, the perceived cost of using a learning strategy has not been fully explored, although the cost of attenuating strategy use seems to be important in considering the use of learning strategies. Therefore, the present research addresses the perceived cost of using a learning strategy in examining the within-person variance-covariance (i.e., we examine variations in the degree of use and subjective cost among multiple strategies within a learner, as well as the within-person relationship between use of learning strategy and perceived cost).

The framework addressed in the present research in assessments of learning strategy use, which typically focuses on within-individual variance, has some methodological advantages. One method to determine the learning strategies used by learners is utilizing a questionnaire approach. In general, such self-assessments include noise such as individual response tendencies and moods, and the measured responses may differ from third party impressions or evaluations based on objective measurements. For example, if there are two respondents, A and B, even if B objectively used a learning strategy k more often than A, if A tended to overestimate the strategy use or B did not happen to use learning strategy k in the task immediately prior to the questionnaire response, their subjective ratings could be reversed in terms of their propensity to use the strategy. Some objective evaluation-based methods have been proposed to avoid such noise (addressed in General Discussion) [24]. The effects of noise, such as mood and evaluation characteristics, can vary in strength between individuals while responding to a questionnaire item with a high or low score. However, even if the measurement is based on self-ratings, the variance between strategies (i.e., within individuals) is likely to be less influenced by a participant's evaluative tendencies or mood at the time than between individuals; this is because the individual's evaluation traits and moods are not expected to bring different noise to the strategies, but to affect the evaluation behavior itself (although the most recent learning or learning behavior, such as the impression of the strategy used at that time, will be noise for the within-person as well as between-person variances). Therefore, we used self-assessment, as is done in many within-participant experiments, to reduce the effects of noise.

The present research focused on the within-individual variance/correlation in the use of multiple learning strategies and addressed perceived costs related to the use of each strategy. To show the within-person variance of the learning strategies, we conducted two studies. We measured cognitive strategies directed related to learning content acquisition in Study 1 and metacognitive strategies in the framework of self-regulated learning in Study 2.

To show the intra-individual correlations between using a strategy and its perceived costs, we must consider that the intentional use of a strategy requires the learner to have knowledge of that strategy and its costs and benefits [13]. We also considered conditional knowledge wherein perceived costs and benefits may change [30]. Since the purpose of this study was to examine the correlation between the use of learning strategies and their cost perceptions within individuals, undergraduate students who were considered to have sufficient knowledge about strategy were selected to participate.

This research consisted of two studies using surveys completed by college students to measure their use of cognitive and metacognitive learning strategies. The survey questions involved perceived cost per condition (when and how) and perceived benefit, knowledge, and utilization of various strategies to measure the use of the strategies and within-person variance-covariance.

The research was approved by the ethical review board of the Psychology department, Hosei University, on March 22, 2013 (there was no approval number system). The ethical considerations were explained to the participants both verbally and in writing prior to initiation of the survey. All data and analyses files are available from the Open Science Framework (OSF) project (https://osf.io/kunpg/).

## Study 1

Study 1 will measure cognitive strategies. As in many previous studies measuring cognitive strategies, we measured deep and shallow processing strategies with reference to the processing level effect in memory research [7]. Measuring different strategies allows for variance between strategies within individuals.

## Methods

**Participants and procedure.** One hundred sixty-three undergraduate students from a Japanese university in the Kanto area agreed to the research objectives and ethical considerations and voluntarily participated in the study. Questionnaires were administered during one of their scheduled classes (educational psychology), for which permission to conduct the survey was obtained from the class instructor (i.e., data of all Study 1 participants were obtained in a one-time survey). The present author conducted the survey, but there was no acquaintance between the participants and researcher. The purpose of the study was explained to all participants who then provided informed consent in writing and verbally. As the number of participants was dependent on the number of students from the class who agreed to participate, the sample sizes were randomly determined; the study did not use a data-collection cessation rule. Participants were excluded from the analysis based on the following three considerations: (a) participants whose survey responses contained more than 20% (i.e., 12 of 56 answer items) or more errors such as non-response to a question, response to more than one numerical value for an item, or no response on a numerical value, (b) participants who were consistently non-responsive to one/more variable(s) in all eight items of the learning strategy (e.g., one of the excluded participants did not respond to all eight learning strategies, only to the question knowledge regarding strategy: *Knowledge* "whether you '[A] were aware before' or '[B] were not aware until hearing about it on this questionnaire' of the existence of the method being asked about"), and (c) participants who gave the same response to all questions during the survey. Finally, 151 students' answers were used for the analysis (34 women; $M_{age}$ = 18.68, $SD_{age}$ = 1.11, $Range_{age}$ = 17–28), except for 12 students. The data for analysis included answers from 151 students regarding eight strategy items for a total of 1,208 responses to Study 1, including those with missing values.

**Measures.** Cognitive strategies used to process information in studying material were measured in an original survey 1 with eight items: Four items were strategies for deep information processing (e.g., "Memorize the process and big picture of what you learned"), and the other four items were strategies for shallow information processing (e.g., "Prioritize memorization over thinking about why it is that way") [31]. The items on cognitive strategy in Study 1 are as follows (as the survey instructions and questionnaire items are in Japanese [31] http://doi.org/10.15002/00010874, the author asked an English editing service [editage]) to perform back-translation to check the accuracy of the translation):

- Memorize after thinking about what the material you were taught means (1) [Memorize Meaning]

- Understand the relationships between various terms (4) [Understand Relationships]

- Memorize the process and big picture of what you learned (5) [Memorize the Big Picture]

- Try to understand the terms you don't know the meaning of as much as possible (8) [Try to Understand]

- Prioritize memorization over thinking about why it is that way (2) [Ignore Origin]

- If you encounter a term you don't know the meaning of, memorize it anyway (3) [Ignore Meaning]

- Begin by memorizing terms before understanding the whole picture (6) [Ignore the Big Picture]

**Table 1. The mean and standard deviation (enclosed in parentheses) of each assessment item of cognitive strategies and the number of people with and without prior strategy knowledge in Study 1.**

| | Memorize Meaning | Understand Relationships | Memorize the Big Picture | Try to Understand | Ignore Origin | Ignore Meaning | Ignore the Big Picture | Just Write |
|---|---|---|---|---|---|---|---|---|
| *Used* | 4.43 | 4.11 | 4.40 | 4.44 | 3.11 | 3.07 | 2.99 | 2.63 |
| | (1.15) | (1.16) | (1.08) | (0.94) | (1.29) | (1.20) | (1.31) | (1.07) |
| *Cost$_{SI}$* | 3.19 | 3.25 | 2.99 | 3.41 | 3.65 | 3.73 | 3.86 | 3.87 |
| | (1.27) | (1.25) | (1.28) | (1.22) | (1.40) | (1.33) | (1.27) | (1.24) |
| *Cost$_{SS}$* | 2.75 | 2.86 | 2.72 | 2.93 | 3.22 | 3.41 | 3.47 | 3.50 |
| | (1.26) | (1.13) | (1.10) | (1.13) | (1.37) | (1.30) | (1.30) | (1.23) |
| *Cost$_{LI}$* | 3.08 | 3.15 | 2.95 | 3.23 | 3.70 | 3.73 | 3.87 | 3.91 |
| | (1.27) | (1.23) | (1.23) | (1.17) | (1.40) | (1.28) | (1.19) | (1.23) |
| *Cost$_{LS}$* | 2.64 | 2.85 | 2.67 | 2.85 | 3.26 | 3.56 | 3.54 | 3.61 |
| | (1.24) | (1.15) | (1.14) | (1.09) | (1.35) | (1.31) | (1.27) | (1.28) |
| *Benefit* | 4.77 | 4.64 | 4.70 | 4.69 | 3.16 | 3.33 | 3.38 | 3.00 |
| | (1.08) | (1.09) | (1.03) | (0.93) | (1.37) | (1.31) | (1.35) | (1.27) |
| Knowledge[a] | 134 | 126 | 133 | 136 | 126 | 117 | 116 | 78 |
| | 12 | 17 | 12 | 9 | 19 | 26 | 28 | 64 |

*Used*, *Cost$_{SI}$*, *Cost$_{SS}$*, *Cost$_{LI}$*, *Cost$_{LS}$* and *Benefit* were all scored in the range of 1 to 6.

[a] The number at the top of the cell indicates the number of respondents who indicated they knew about the strategy (coded as 0.5), while the number at the bottom of the cell indicates the number of respondents who indicated they did not know about the strategy (coded as -0.5).

- Try to be able to accurately write terms rather than understand their meaning (7) [Just Write]

The order in which the items are presented here corresponds to Table 1. Numbers in parentheses after the description indicate the items' order of presentation in the questionnaire, whereas the labels in square brackets match the headings in Table 1. Questionnaires were randomly distributed in reverse order due to counterbalancing.

Participants were required to respond to seven items for each strategy. The teaching and response items presented to the participants were as follows:

The following items indicate ways of studying that you may or may not use in your learning process. For each item, please circle a number for the following six: ① How well does this apply to your current way of studying? ② How cumbersome do you think it would be to use this method at all times, regardless of the circumstances, in order to improve your score on your next test? ③ How cumbersome do you think it would be to use this method only when it suits the situation in order to score points on your next test? ④ How cumbersome do you think it would be to use this method at all times, regardless of the circumstances, in continuing your future learning? ⑤ How cumbersome do you think it would be to use this method only when it suits the situation in continuing your future learning? ⑥ How effective do you think this method is? After "How well does this apply to your current way of studying?" for a given method, you will be asked ⑦ whether you "(A) were aware of this strategy beforehand" or "(B) were not aware about this strategy until reading about it on this questionnaire." Please circle the corresponding letter. There are no right or wrong answers, so please answer based on what you think is accurate.

1. Actually using

2. Using at all times in order to score points on my next test would be cumbersome

3. Using only when it fits the situation in order to score points on my next test would be cumbersome

4. Using at all times in continuing my future learning would be cumbersome

5. Using only when it fits the situation in continuing my future learning would be cumbersome

6. Effective to use

7. I (A) was aware of (B) wasn't aware of the existence of this method

Four of the seven strategy aspects were perceived costs, consisting of a combination of time perspective (short, long) and beliefs about strategy use (irrelevant, suitable) [26]. An example of a short-term perspective is "this strategy works well when used to learn something quickly," and an example of a long-term perspective is "works for important information that needs to be retained for future use and built upon for further learning." The belief about strategy reflects a sense of self-regulation, such choosing the strategy according to the task. Including the four perceived costs, the seven aspects are as follows: usage of strategy, knowledge of strategy, perceived short-irrelevant cost, perceived short-suitable cost, perceived long-irrelevant cost, perceived long-suitable cost, and perceived benefit. Knowledge regarding strategy was binary ("knew of," coded as 0.5, or "did not know of," coded as -0.5), whereas the other aspects were scored on a Likert scale ranging from 1 ("Not true at all") to 6 ("Very true"). Participants were asked for responses to eight items in each of seven aspects, a total of 56 items. Descriptive statistics, mean and standard deviation of learning strategy use (*Used*), perceived short-irrelevant cost ($Cost_{SI}$), perceived short-suitable cost ($Cost_{SS}$), perceived long-irrelevant cost ($Cost_{LI}$), perceived long-suitable cost ($Cost_{LS}$), and perceived benefit (*Benefit*) as well as a summary of participants' strategy knowledge (*Knowledge*), are shown in Table 1.

**Data structure, modeling, and analysis.** Intra-individual variance and correlation were calculated to examine the relationship between the individual learner's non-use strategy and the perceived cost. The data set was organized as follows: A convenient number was assigned to each strategy, and one strategy item variable was input on one line. The participants in survey 1 answered eight items about cognitive strategies, so data per person were spread over eight lines. The structure of such data was the same as person-period data in the longitudinal research, but it is different in that all participants respond to the same item, so the analysis result includes the unique effect that each strategy item has on the variable of the strategy. The effect of such items that are not research interests but may influence the analysis results is called the random item effect, and type I error rates have been found to increase unjustly when random-item effects are ignored [32]. The difference between the results of the participants, that is, the individual differences, is included in the analysis results as the random participant effect, and it is distinguished from the random item effect. The data-point (DP) used for analysis obtained in Study 1 was 151 students answers to eight strategy items for a total of 1,208 responses for each aspect, including missing values.

To appropriately estimate random-participant and random-item effects, a mixed-effects model was used for these data [33]. There are some noteworthy points when building a model that takes random effects into account. Whether it is a random-participant or random-item effect, it is necessary to establish whether the differences in participants or items affect the scores of the use of the learning strategy (i.e., the random intercept), the relationship between the explanatory and objective variables (i.e., the random slope), or both. In addition, it is possible to separately set random effects for the participants and items, such as only the intercept for the random-participant effect, or both the intercept and the slope for the random-item

effect. As mentioned earlier, this study does not have a rationale for setting random effects because it examines the within-individual variance (between-item variance) in learning strategy research. Thus, because of the exploratory stance of this study toward random effects, we decided to use the maximal model with random effects of intercept and slope in both the random-participant and random-item effects [34]. First, the intra-class correlation (ICC) coefficient among participants was calculated for each strategy variable without entering independent variables. The equation for ICC was as follows:

$$Used_{ij} = \beta_{0ij} + r_{ij} \tag{1}$$

$$\beta_{0ij} = \gamma_{00} + \mu_{0j} + v_{0i} \tag{2}$$

$$Var(\mu_{0j}) = \tau_{00} \tag{3}$$

$$Var(v_{0i}) = \omega_{00} \tag{4}$$

where the outcome variable, $Used_{ij}$, represents the amount of learning strategy use for item $i$ and participant $j$. Parameter $r_{ij}$ is the error term, assuming normal distribution, at the within-person level. $\beta_{0ij}$ in Eq 1 is the intercept that defines three parameters in Eq 2: $\gamma_{00}$, which is between-person intercept, $\mu_{0j}$ which is participant residual error for participant $j$, and $v_{0i}$ which is item residual error for item $i$. Parameters $\tau_{00}$ and $\omega_{00}$ indicate the variance of participants and items residual error, respectively. Second, independent variables were added to Eqs 1 and 2.

$$
\begin{aligned}
Used_{ij} \quad &= \beta_{0ij} \\
&+ \beta_{1ij}(Cost_{SI})_{ij} + \beta_{2ij}(Cost_{SS})_{ij} + \beta_{3ij}(Cost_{LI})_{ij} + \beta_{4ij}(Cost_{LS})_{ij} \\
&+ \beta_{5ij}(Benefit)_{ij} + \beta_{6ij}(Knowledge)_{ij} \\
&+ r_{ij}
\end{aligned}
\tag{5}
$$

$$\beta_{1ij} = \gamma_{10} + \mu_{1j} + v_{1i} \tag{6}$$

$$\beta_{2ij} = \gamma_{20} + \mu_{2j} + v_{2i} \tag{7}$$

$$\beta_{3ij} = \gamma_{30} + \mu_{3j} + v_{3i} \tag{8}$$

$$\beta_{4ij} = \gamma_{40} + \mu_{4j} + v_{4i} \tag{9}$$

$$\beta_{5ij} = \gamma_{50} + \mu_{5j} + v_{5i} \tag{10}$$

$$\beta_{6ij} = \gamma_{60} + \mu_{6j} + v_{6i} \tag{11}$$

$$Var(\mu_{1j}) = \tau_{11},$$

$$Var(\mu_{2j}) = \tau_{22},$$

$$Var(\mu_{3j}) = \tau_{33},$$

$$Var(\mu_{4j}) = \tau_{44},$$  \qquad (12)

$$Var(\mu_{5j}) = \tau_{55},$$

$$Var(\mu_{6j}) = \tau_{66}.$$

$$Var(v_{1i}) = \omega_{11},$$

$$Var(v_{2i}) = \omega_{22},$$

$$Var(v_{3i}) = \omega_{33},$$

$$Var(v_{4i}) = \omega_{44},$$  \qquad (13)

$$Var(v_{5i}) = \omega_{55},$$

$$Var(v_{6i}) = \omega_{66}.$$

where the six parameters, $\beta_{1ij}$ to $\beta_{6ij}$, represent the linear effects of perceived short-irrelevant cost (*Cost$_{SI}$*), perceived short-suitable cost (*Cost$_{SS}$*), perceived long-irrelevant cost (*Cost$_{LI}$*), perceived long-suitable cost (*Cost$_{LS}$*), perceived benefit (*Benefit*), and knowledge regarding strategy (*Knowledge*) on usage of learning strategy (*Used*), respectively, for participant $j$ and item $i$. Furthermore, the six parameters consist of the mean of slope across participants $\gamma_{10}$ to $\gamma_{60}$ (i.e., intercept of $\beta$), between person residual error $\mu_{1j}$ to $\mu_{6j}$ (i.e., variance $\tau_{11}$, $\tau_{22}$, $\tau_{33}$, $\tau_{44}$, $\tau_{55}$ and $\tau_{66}$), and between item residual error $v_{1i}$ to $v_{6i}$ (i.e., variance $\omega_{11}$, $\omega_{22}$, $\omega_{33}$, $\omega_{44}$, $\omega_{55}$ and $\omega_{66}$), respectively.

As shown, there are many parameters to be estimated, and it is necessary to assume normality for each parameter. Since it is difficult to keep to the assumption of normality, a Markov chain Monte Carlo (MCMC) method was used to estimate the interval which included the parameter (i.e., credible interval), namely the Bayesian hierarchical model [35]. The MCMC method is an algorithm for sampling probability distributions using random numbers, and it can be used instead of the maximum likelihood estimation method to obtain estimates and their intervals. The estimates presented in this study report the median of the posterior distribution. In addition, this study presents a credible interval, which is the range in which the population is included with 95% probability, as an alternative to the confidence interval required by the maximum likelihood estimation method; additionally, it refers to the relationship between the explanatory and objective variables based on whether or not the credible interval contains zero. The calculation involved a minimum of 10,000 iterations, and the first half of each chain was discarded as part of the burn-in phase in Mplus ver. 8.3 [36]. The MCMC algorithm was repeated four times with different initial values. The seed value was set to 1015. For the analysis, variable transformation and missing value analysis were performed. First, knowledge regarding strategies ([*Knowledge*]$_{ij}$)was coded as 0.5 (knew about the strategy) or -0.5 (did not know about the strategy). Second, the multiple imputation method was used as a treatment for missing values. After considering the crossed data structure of the strategy items and participants, we substituted the regression prediction plus the residuals for the missing data points. This procedure was repeated 20 times to create 20 data sets with different assigned

values, and the analysis was performed on each data set. The integrated results of the 20 analyses are reported in this study. Third, an analysis of the mixed-effects model was performed for each data set created by the multiple imputation method, and the results were integrated. At that time, to properly separate between-person variance from within-person variance, perceived costs, and benefit ($[Cost_{SI}]_{ij}$, $[Cost_{SS}]_{ij}$, $[Cost_{LI}]_{ij}$, $[Cost_{LS}]_{ij}$, and $[Benefit]_{ij}$) were centered within individuals (for centering within a participant [37]).

## Results

**Intraclass and within-person correlations.** Table 2 shows the intra-class correlation ($ICC$) and within-person correlation coefficients. First, $ICC$ (italicized in Table 2) were generally low ($ICCs < .27$). The use of learning strategies was particularly low in its value, with a large proportion of intra- and inter-individual-strategy variance ($ICC_{Used} = .051$). Next, for within-person correlations, the learning strategy use and the four perceived costs were each negatively valued ($-.53 < rs_{Used-Costs} < -.48$), and positively valued for perceived benefit and knowledge about strategy ($r_{Used-Benefit} = .69$, $r_{Used-Knowledge} = .34$). There was also a high correlation between the four perceived costs ($rs > .64$).

**Within-person relationship between perceived costs and learning strategy use.** Table 3 shows the results of the Bayesian hierarchical modeling analysis with the use of learning strategies as the dependent variable and the four perceived costs, perceived benefit, and knowledge about strategy as independent variables. A direct effect of attenuating the use of learning strategy was shown to be the perceived cost of continuing to use it for the next test ($\gamma_{10} = -0.16$ [$-0.30 - -0.03$]). There was also a relationship between the strategies with a high perceived benefit and those that the individuals have knowledge about, such that they were used more frequently ($\gamma_{50} = 0.35[0.21-0.43]$, $\gamma_{60} = 0.61[0.29-0.99]$). Note that except for individual differences in the use of strategies and the presence or absence of knowledge about strategy ($\tau_{00} = 0.20[0.13-0.27]$, $\tau_{66} = 0.35[0.17-0.67]$), the variance between individuals ($\tau_{11} - \tau_{55}$), and the variance between strategies, ($\omega_{00} - \omega_{66}$) was small. When the variance inflation factor (VIF) between explanatory variables was calculated from the within-person correlation coefficients, the highest correlation coefficient was .81, with a VIF of 2.91, indicating between $Cost_{LI}$ and $Cost_{LS}$.

## Discussion

Study 1 showed within-individual correlations between the use of cognitive strategies and perceived costs. The $ICC$ coefficients resulted in a greater proportion of within-individual variance than between-individual variance. This result indicates that the degree of use varies from strategy to strategy. Moreover, this within-individual variance in strategy use was explained not only by the perceived benefit shown in previous studies, but also by perceived cost.

**Table 2. The intra-class correlation coefficients per item of cognitive strategies (diagonal italic type) and within-person correlation coefficients in Study 1.**

| | Variable | 1 | 2 | 3 | 4 | 5 | 6 | 7 |
|---|---|---|---|---|---|---|---|---|
| 1 | *Used* | *.05* | | | | | | |
| 2 | $Cost_{SI}$ | -.48 | *.27* | | | | | |
| 3 | $Cost_{SS}$ | -.51 | .73 | *.27* | | | | |
| 4 | $Cost_{LI}$ | -.49 | .79 | .66 | *.24* | | | |
| 5 | $Cost_{LS}$ | -.53 | .64 | .81 | .77 | *.24* | | |
| 6 | *Benefit* | .69 | -.36 | -.44 | -.42 | -.47 | *.07* | |
| 7 | *Knowledge* | .34 | -.18 | -.21 | -.21 | -.23 | .26 | *.25* |

**Table 3. Results of Bayesian hierarchical modeling with the use of cognitive strategies as the dependent variable and participants and strategy items as random effects in Study 1.**

| | Variable | Estimation | | |
|---|---|---|---|---|
| | | *Lower* | *Median* | *Upper* |
| | | Coefficient | | |
| $\gamma_{00}$ | *(Intercept)* | 3.18 | 3.42 | 3.69 |
| $\gamma_{10}$ | $Cost_{SI}$ | -0.30 | -0.16 | -0.03 |
| $\gamma_{20}$ | $Cost_{SS}$ | -0.30 | -0.13 | 0.05 |
| $\gamma_{30}$ | $Cost_{LI}$ | -0.17 | -0.02 | 0.13 |
| $\gamma_{40}$ | $Cost_{LS}$ | -0.22 | -0.07 | 0.04 |
| $\gamma_{50}$ | *Benefit* | 0.21 | 0.35 | 0.43 |
| $\gamma_{60}$ | *Knowledge* | 0.29 | 0.61 | 0.99 |
| $r_{ij}$ | *(residual)* | 0.42 | 0.47 | 0.52 |
| | | Variances by Participants | | |
| $\tau_{00}$ | *(Intercept)* | 0.13 | 0.20 | 0.27 |
| $\tau_{11}$ | $Cost_{SI}$ | 0.00 | 0.01 | 0.04 |
| $\tau_{22}$ | $Cost_{SS}$ | 0.05 | 0.07 | 0.16 |
| $\tau_{33}$ | $Cost_{LI}$ | 0.00 | 0.01 | 0.03 |
| $\tau_{44}$ | $Cost_{LS}$ | 0.00 | 0.02 | 0.06 |
| $\tau_{55}$ | *Benefit* | 0.04 | 0.07 | 0.11 |
| $\tau_{66}$ | *Knowledge* | 0.17 | 0.35 | 0.67 |
| | | Variances by Items | | |
| $\omega_{00}$ | *(Intercept)* | 0.02 | 0.09 | 0.46 |
| $\omega_{11}$ | $Cost_{SI}$ | 0.00 | 0.01 | 0.09 |
| $\omega_{22}$ | $Cost_{SS}$ | 0.00 | 0.02 | 0.16 |
| $\omega_{33}$ | $Cost_{LI}$ | 0.00 | 0.02 | 0.12 |
| $\omega_{44}$ | $Cost_{LS}$ | 0.00 | 0.01 | 0.07 |
| $\omega_{55}$ | *Benefit* | 0.00 | 0.00 | 0.05 |
| $\omega_{66}$ | *Knowledge* | 0.01 | 0.10 | 0.71 |

"*Lower*" and "*Upper*" denote the lower and upper limits of the credible interval for "Estimation", respectively.

Concretely, it was shown that they were not used as much as cognitive strategies that are considered cumbersome to be used over a short period of time. Participants are more likely to associate cognitive strategies like memorization and comprehension with their current studying because the strategies are directly related to the acquisition of learning content. Therefore, a short timeframe (e.g., the next test) and a condition in which participants did not have to consider the appropriate use of the cognitive strategy (i.e., they did not take into account the circumstances) may have predicted the reduced use of the cognitive strategy. In other words, it is possible that the frequency of cognitive strategy use decreases only in the "short-term" and "irrelevant situation" conditions in the subjective perceived cost held by the individual learner, but not in the other conditions. However, the within-individual correlations between the four perceived costs were high, which suggests that they may have an effect on inhibiting the use of learning strategies, although the conditions were different. In other words, referring to previous studies, we also established 2 (when: short-term OR long-term) x 2 (how: not considering situation OR fits situation) conditional knowledge about perceived cost for the use of learning strategies [6, 25, 26]; however, the between-strategy correlation coefficients among the four variables were high (to the extent that we did not have to worry about multicollinearity), and it

is possible that learners do not discriminate between strategies measured in Study 1 least and conditional knowledge such as "when" and "how." Thus, the results of this study illustrate that we can examine the differences in the degree of use and the determinants of the degree of use between the strategies within individuals, even if measured in the same way as the conventional method. Such findings of within-individual variance and covariance may be useful in promoting the use of learning strategies since they more directly capture the process of strategy use.

## Study 2

Study 2 will measure metacognitive strategies. We measured monitoring and control strategies for metacognitive activity [8], so that the same variance between strategies occurs within individuals as for cognitive strategies. Monitoring is an activity in which the learner reflects on and evaluates their own learning situation and past behavior, while control is an activity in which the learner reviews their own learning behavior and future plans in response to monitoring. As it is desirable for metacognitive activities to activate both of these activities, we measured both activities so that we can examine which one is not being used.

### Methods

**Participants and procedure.** One hundred eighty-eight undergraduate students, different than the group of participants in Study 1, from a Japanese university in the Kanto area agreed with the research objectives and ethical considerations and voluntarily participated in Study 2. The questionnaires were administered in a class. The purpose of the study was explained to all participants, who provided informed consent was provided in writing and verbally. As the number of participants depended on the number of students from class who agreed to participate, sample size was randomly determined and the study did not use a data-collection cessation rule. Participants were excluded from the analysis based on the following three considerations: (a) participants whose survey responses contained more than 20% (i.e., 10 of 49 answer items) or more errors such as non-response to a question item, response to more than one numerical value for an item, or no response on a numerical value, (b) participants who were consistently non-responsive to one/more variable(s) in all seven items of the learning strategy, and (c) participants who gave the same response to all questions during the survey. Finally, 158 students' answers were used for analysis (92 women; $M_{age}$ = 19.25, $SD_{age}$ = 1.90, $Range_{age}$ = 18–38, Eight participants had unknown age), except for 30 students. The data for analysis included answers from 158 students regarding seven strategy items for a total of 1,106 responses in to study 2, including those with missing values.

**Measures.** Metacognitive strategies used to monitor and control one's own learning behavior were measured in the survey in Study 2 [26, 31]. Seven items were measured: four items for strategies for monitoring one's own learning activities (e.g., "Predict how much you will probably be able to accomplish"), and other three items about strategies for controlling one's own learning behavior (e.g., "Try to use an appropriate study method for each topic"). The items of the metacognitive strategy in Study 2 are as follows:

- Predict how much you will probably be able to accomplish (1) [Predict Accomplishment]

- Try to clarify how well you understand the content (3) [Clarify Understanding]

- Try to grasp your level of understanding (4) [Grasp Understanding]

- Try to evaluate the extent of your understanding (6) [Evaluate Understanding]

- Try to plan how to study first (2) [Study Planning]

**Table 4. The mean and standard deviation (enclosed in parentheses) of each assessment item of metacognitive strategies and the number of people with and without prior strategy knowledge in Study 2.**

| | Predict Accomplishment | Clarify Understanding | Grasp Understanding | Evaluate Understanding | Study Planning | Appropriate Studying | Set Achievement |
|---|---|---|---|---|---|---|---|
| *Used* | 3.41 | 4.01 | 4.12 | 3.30 | 3.83 | 4.15 | 3.74 |
| | (1.30) | (1.15) | (1.12) | (1.25) | (1.37) | (1.20) | (1.36) |
| $Cost_{SI}$ | 3.22 | 3.08 | 3.11 | 3.39 | 3.53 | 3.15 | 3.25 |
| | (1.22) | (1.29) | (1.29) | (1.31) | (1.48) | (1.27) | (1.27) |
| $Cost_{SS}$ | 3.04 | 2.99 | 2.98 | 3.24 | 3.27 | 2.93 | 2.96 |
| | (1.13) | (1.22) | (1.20) | (1.23) | (1.37) | (1.17) | (1.13) |
| $Cost_{LI}$ | 3.26 | 3.08 | 3.03 | 3.29 | 3.42 | 3.05 | 3.06 |
| | (1.20) | (1.29) | (1.27) | (1.26) | (1.43) | (1.24) | (1.24) |
| $Cost_{LS}$ | 3.14 | 3.00 | 2.93 | 3.18 | 3.20 | 2.94 | 2.91 |
| | (1.17) | (1.19) | (1.20) | (1.23) | (1.32) | (1.20) | (1.14) |
| *Benefit* | 3.98 | 4.33 | 4.44 | 4.03 | 4.54 | 4.31 | 4.29 |
| | (1.13) | (1.09) | (1.07) | (1.20) | (1.19) | (1.17) | (1.20) |
| *Knowledge*[a] | 93 | 122 | 129 | 93 | 138 | 124 | 116 |
| | 57 | 26 | 19 | 55 | 9 | 25 | 34 |

*Used*, $Cost_{SI}$, $Cost_{SS}$, $Cost_{LI}$, $Cost_{LS}$ and *Benefit* were all scored in the range of 1 to 6.

[a] The number at the top of the cell denotes the number of respondents who answered indicated they knew about the strategy (coded as 0.5), while the number at the bottom of the cell indicates the number of respondents who indicated they did not know about the strategy (coded as -0.5).

- Try to use an appropriate study method for each topic (5) [Appropriate Studying]

- Try to set a goal for how much you aim to achieve (7) [Set Achievement]

The order in which the items are presented here corresponds to Table 4. Numbers in parentheses after the description indicate the order of presentation in the questionnaire, whereas the labels in square brackets match the headings in Table 4. Questionnaires were also randomly distributed in reverse order due to counterbalancing.

Participants were required to respond to the following seven aspects of each strategy item, as in Study 1: usage of strategy, knowledge regarding strategy, perceived short-anyway cost, perceived short-suitable cost, perceived long-anyway cost, perceived long-suitable cost, and perceived benefit. Ultimately, participants were asked for responses to seven items for each of seven aspects for a total of 49 items. Descriptive statistics, mean and standard deviation of *Used*, $Cost_{SI}$, $Cost_{SS}$, $Cost_{LI}$, $Cost_{LS}$, and *Benefit*, and a summary of participants' *Knowledge* are shown in Table 4. The structure of the data, the modeling used for analysis, and other procedures for analysis were all the same as for Study 1.

## Result and discussion

**Intraclass and within-person correlations.** Table 5 shows the ICC and within-personal correlation coefficients. First, mostly moderate values for the ICC coefficients were shown (.23 < *ICCs* <.57). Next, for intrapersonal correlations, the use of learning strategies and the four perceived costs were each negatively valued ($-.25 < rs_{Used-Costs} < -.17$), while perceived benefit and knowledge about strategy were positively valued ($r_{Used-Benefit} = .38$, $r_{Used-Knowledge} = .37$). There was also a reasonably high correlation between the four cost sensitivities (.58 < *rs* <.75).

**Within-person relationship between perceived costs and learning strategy use.** As in Study 1, the results of the analysis using Bayesian hierarchical modeling with *Used* as the

**Table 5. The intra-class correlation coefficients per item of metacognitive strategies (diagonal italic type) and within-person correlation coefficients in Study 2.**

| | Variable | 1 | 2 | 3 | 4 | 5 | 6 | 7 |
|---|---|---|---|---|---|---|---|---|
| 1 | *Used* | *.29* | | | | | | |
| 2 | $Cost_{SI}$ | -.22 | *.54* | | | | | |
| 3 | $Cost_{SS}$ | -.25 | .65 | *.52* | | | | |
| 4 | $Cost_{LI}$ | -.19 | .69 | .60 | *.57* | | | |
| 5 | $Cost_{LS}$ | -.17 | .58 | .75 | .69 | *.53* | | |
| 6 | *Benefit* | .38 | -.02 | -.04 | -.02 | -.05 | *.43* | |
| 7 | *Knowledge* | .37 | -.05 | -.04 | -.04 | -.05 | .21 | *.23* |

dependent variable and $Cost_{SI}$, $Cost_{SS}$, $Cost_{LI}$, $Cost_{LS}$, *Benefit*, and *Knowledge* as independent variables are shown in Table 6. First, for the effect of the four perceived costs, $Cost_{SS}$, "[this strategy] is cumbersome to use properly for the next exam," suppressed learning strategy use ($\gamma_{20} = -0.208[-0.365 - -0.038]$). Also, the strategies with higher perceived benefit and those with more knowledge were used more frequently ($\gamma_{50} = 0.325[0.180 - 0.466]$, $\gamma_{60} = 1.022[0.737 - 1.362]$). As in Study 1, except for individual differences in the use of strategies and the presence

**Table 6. Results of Bayesian hierarchical modeling with the use of metacognitive strategies as the dependent variable and participants and strategy items as random effects in Study 2.**

| | Variable | Estimation | | |
|---|---|---|---|---|
| | | *Lower* | *Median* | *Upper* |
| | | Coefficient | | |
| $\gamma_{00}$ | (Intercept) | 3.31 | 3.51 | 3.71 |
| $\gamma_{10}$ | $Cost_{SI}$ | -0.30 | -0.11 | 0.05 |
| $\gamma_{20}$ | $Cost_{SS}$ | -0.37 | -0.21 | -0.04 |
| $\gamma_{30}$ | $Cost_{LI}$ | -0.28 | -0.08 | 0.09 |
| $\gamma_{40}$ | $Cost_{LS}$ | -0.08 | 0.13 | 0.36 |
| $\gamma_{50}$ | Benefit | 0.18 | 0.33 | 0.47 |
| $\gamma_{60}$ | Knowledge | 0.74 | 1.03 | 1.36 |
| $r_{ij}$ | (residual) | 0.47 | 0.51 | 0.59 |
| | | Variances by Participants | | |
| $\tau_{00}$ | (Intercept) | 0.29 | 0.41 | 0.55 |
| $\tau_{11}$ | $Cost_{SI}$ | 0.01 | 0.05 | 0.12 |
| $\tau_{22}$ | $Cost_{SS}$ | 0.00 | 0.02 | 0.07 |
| $\tau_{33}$ | $Cost_{LI}$ | 0.00 | 0.03 | 0.08 |
| $\tau_{44}$ | $Cost_{LS}$ | 0.02 | 0.10 | 0.18 |
| $\tau_{55}$ | Benefit | 0.06 | 0.13 | 0.21 |
| $\tau_{66}$ | Knowledge | 0.16 | 0.46 | 0.75 |
| | | Variances by Items | | |
| $\omega_{00}$ | (Intercept) | 0.00 | 0.02 | 0.20 |
| $\omega_{11}$ | $Cost_{SI}$ | 0.00 | 0.01 | 0.18 |
| $\omega_{22}$ | $Cost_{SS}$ | 0.00 | 0.01 | 0.15 |
| $\omega_{33}$ | $Cost_{LI}$ | 0.00 | 0.02 | 0.19 |
| $\omega_{44}$ | $Cost_{LS}$ | 0.00 | 0.02 | 0.28 |
| $\omega_{55}$ | Benefit | 0.00 | 0.01 | 0.10 |
| $\omega_{66}$ | Knowledge | 0.01 | 0.07 | 0.59 |

"*Lower*" and "*Upper*" denote the lower and upper limits of the credible interval for "Estimation", respectively.

or absence of knowledge about strategy ($\tau_{00} = 0.409[0.293–0.548]$, $\tau_{66} = 0.460[0.163–0.746]$), the variance between individuals ($\tau_{11} – \tau_{55}$) and the variance between strategies ($\omega_{00} – \omega_{66}$) was small. When the VIF between explanatory variables was calculated from the within-person correlation coefficients, the highest correlation coefficient was .75, with a VIF of 2.25, indicating between $Cost_{LI}$ and $Cost_{LS}$ (i.e., no concern for multicollinearity in Study 2, as in Study 1).

Study 2 showed the within-individual correlation between the use of metacognitive strategies and perceived cost. The relationships shown by the within-person correlations and the Bayesian hierarchical modeling were generally consistent with cognitive strategies. On the other hand, the *ICC* coefficient showed that the proportion of inter-individual variance in the use of metacognitive strategies was higher than for cognitive strategies, and the number of within-individual correlations was lower. Bayesian hierarchical modeling showed that a perceived cost, which differs in conditional knowledge from cognitive strategies, directly explained the use of the strategies: Although the conditions of time perspective, such as learning for the next test, were the same as those of the cognitive strategies, the conditions of usage were different, such as appropriate use of the strategy ($\gamma_{10}[Cost_{SI}] = −0.16$ in Study 1, whereas $\gamma_{20}[Cost_{SS}] = −0.21$ in Study 2; since both of these did not include 0 in the 95% confidence interval, it is likely that there is a negative effect on the use of learning strategies). In Study 2, only $Cost_{SS}$ predicted to *Used*, and the rest of $Cost_{SI}$, $Cost_{LI}$, and $Cost_{LS}$ were determined not to predict strategy use because the 95% Bayesian credible interval contained 0. Metacognitive strategies have a role in controlling learning behavior rather than being involved in learning content. As with the cognitive strategies, the distant future condition does not predict the frequency of metacognitive strategy use, and it is possible that learners perceive studying as something to be done in the near future to show results on a test, both in terms of perceived benefit [25] and cost. It is interesting to note that unlike cognitive strategies, the conditions of use appropriate to the situation discouraged the use of metacognitive strategies. Metacognition, as the term implies, is a higher-order cognition of the target cognition/behavior, and thus may be difficult for participants to evaluate. Therefore, metacognitive strategies are difficult to use, and the results of this study (i.e., $Cost_{SS}$ predicting *Used*) may reflect confusion about judging the "appropriate use" of metacognitive strategies in a situation. However, as with cognitive strategies, the discrimination of perceived cost by condition (i.e., time perspective and how to use) may not be apparent to the learner because the within-person correlation was, to some extent, high in the four perceived costs.

## General discussion

The present research illustrates the results of a model and analysis of intra-individual variance-covariance that can reveal the within-person process reflecting the why a learner uses one among multiple strategies. Study 1 measured eight cognitive strategies and showed within-individual variance of strategy use and within-individual correlation between strategy use and perceived cost. Similarly, Study 2 used seven metacognitive strategies to examine intra-individual variance and correlations. Although intra-individual variance and covariance were smaller than cognitive strategies in Study 2, the more cost-sensitive strategies were less likely to be used.

While much of the existing research examining the use of learning strategies has addressed inter-individual variance-covariance, more attention has been paid to intra-individual variation in recent years [23, 24]. In the framework of self-regulated learning, it is desirable for a learner to flexibly change the use of a strategy according to the task and learning content. Therefore, this study focused on variance in the use of multiple strategies by an individual [6]. Our results show that subjective cognition of a strategy, especially the perception of the cost

being that a strategy is a hassle to use, may inhibit the use of the strategy in the present research. Our two studies examined the within-individual variance of cognitive and metacognitive strategies and showed differences in the trends of within-individual variance for each strategy. Specifically, the following differences were suggested between the cognitive and metacognitive strategies within a (single) learner: For cognitive strategies, the higher the subjective cost of a given strategy, the less likely participants were to use it; Conversely, while metacognitive strategies showed the same correlation as cognitive strategies vis-à-vis temporal conditional knowledge (i.e., using a given strategy in the short term, such as in a future test), the higher the cost of a strategy's methodology-related conditional knowledge (using a given strategy if the circumstances are suitable), the less likely were learners to use it. Thus, it was shown that cognitive and metacognitive strategies may differ in their conditional knowledge of costs that lead to variance in the degree of use. However, because of the different participants and their sample sizes, the results of Studies 1 and 2 cannot be directly compared, and the differences in the behavior of within-individual variance between cognitive and metacognitive strategies should be mentioned with data measured on the same sample of these strategies. Although differences in the determinants of cognitive and metacognitive strategies and their impact on academic performance from inter-individual variance and covariance have been addressed in the past, it is important for educational practice that different behaviors for within-individual variance and covariance are known as well.

As mentioned in the Introduction, the within-individual perspective can be interpreted as "The use of y in strategy k for given learner j increases when intra-personal variable x is higher." For example, between-person correlations are interpreted as "the more cumbersome the strategy is, the less likely it is to be used," while within-person correlations are interpreted as "the more cumbersome a strategy is, the less likely it is to be used, compared with less cumbersome strategies." As shown in Fig 2, such a reference is possible due to changes in the data set (i.e., the items of a strategy lie in one row) and the implementation of appropriate analysis methods (such as centering within-person and Bayesian hierarchical modeling), even though the data acquisition data remain the same. The reason for proposing a within-person approach in learning strategies research is that most of the research dealing with learning strategies propose to promote their use. In order to respond more directly to this suggestion, it is possible to change the degree of strategy use by examining whether the degree of use depends on the subjective cognition of the learning strategy that the learners have, and by approaching that subjective cognition, it is possible to change the degree of strategy use. To date, many previous studies have shown that learners may not always perceive theoretically effective learning strategies to be effective [9–12, 18–20]. (In this study, the deep processing strategy in Study 1 [e.g., "Understand the relationships between various terms"] and the metacognitive strategy in Study 2 in general were theoretically effective strategies. As shown in Tables 1 and 4, unlike previous studies, participants in this study seemed to perceive theoretically effective strategies as subjectively effective as well.) Fortunately, there is some evidence that approaching learners about their misperceptions can promote the use of theoretically effective strategies [18]. In the case of inter-individual variance, the main focus was often on the quality and quantity of an individual's motivation and their perception of the task [1, 12]. Traditional between-individual approaches, which attempt to identify the characteristics of individuals who use a strategy frequently, are, of course, important. Equally important, the within-individual approach illustrated in this study provides more direct and practical insight into individual learners' use of various strategies.

We will now summarize the SRL framework and its relationship to the knowledge regarding strategy and subjective cost/benefit of using learning strategies from the within-person perspective presented in this study. The first is the trade-off between cost and benefit in using a

learning strategy. Learners tend to evaluate learning strategies that require less mental effort to use as more effective [38]. In other words, there is a negative correlation between perceived cost and perceived effectiveness. However, such a trade-off is a false belief, and in fact, the use of learning strategies with higher perceived cost has been shown to increase performance. Thus, it is possible that subjective cost/benefit is not unidimensional, but instead, mental effort is related only to perceived cost, and the actual cost/benefit of a learning strategy can be independent or even positively correlated (i.e., the more cumbersome a strategy is to use, the better the learning effect). Second, the present research examined two conditions for the perceived cost: "When is the learning for (for the next exam or for the future?)" and "How do you handle it (use it irrelevant a situation or use it as needed?)". However, these conditions were not based on the actual learning that the learners were doing, but were vague questions that could be answered intuitively. In this regard, Kornell and Finn specifically defined "when" as the beginning and end of study, and "how" as the way of studying [39]. In order to provide study guidance to students, it would be desirable for the learners themselves to be specific about their actions, since it is easier for them to know what to do. On the other hand, this study also reiterated that both the perceived benefit and cost are subjective to the learner, and furthermore, to the frequency of the learning strategies used. Therefore, we believe that asking questions in vague conditions such as the ones in this study may help us pinpoint learners' assumptions, and then we can teach from conditions that can be reflected in specific learning behaviors.

Third, the steps leading up to the use of the learning strategies envisioned in this study. We measured knowledge regarding strategy, procedural knowledge, perceived benefit and cost, and conditional knowledge, referring to the memory and decision-making framework (although we could not measure procedural knowledge on the questionnaire). Although this study showed the process of how learners use learning strategies, similar instructional steps have been proposed by McDaniel and Einstein in terms of training self-regulated learners: (a) acquiring knowledge about strategies, (b) belief that the strategy works, (c) commitment to using the strategy, and (d) planning of strategy implementation [40]. We believe that the steps of this study and theirs are not in conflict or contradiction, but rather complementary relationships. In particular, the commitment and planning process is a unique perspective of teaching and will be essential for establishing the use of learning strategies. In fact, it has been reported that learners who are highly motivated to avoid failure tend to use strategies that are less effective in learning [12]. It is also known that even if the effectiveness of a strategy is often underestimated, the effectiveness of the strategy can be appropriately recognized by teaching the theoretical effectiveness of the strategy to correct the misunderstanding and by experiencing its effectiveness [18]. Instead, *knowledge regarding strategy* (measured in this study) and *procedural knowledge* (only mentioned) are steps that elaborate on their *knowledge about strategy*, and *subjective cost/benefit* is a step that elaborates on their *beliefs*. This study shows that the perceived cost of a learning strategy may reduce its use independent of perceived benefit, and that cost is more strongly reflected in that decision than benefit in the decision-making framework referred to [21, 27, 28]. We also argue that the process of acquiring procedural knowledge is also important, although this is an issue that we have not been able to examine in this study. Simple strategies, such as just repeating words out loud, do not require much practice, and knowing them may be enough, whereas strategies such as memorizing related content together may be cumbersome to execute after memorizing the procedural knowledge. It is known that the striatum is activated in (model-free) reinforcement learning, where trials are repeated to maximize rewards [41], and it is thought that the striatum is also deeply involved in procedural memory [42]. From this point of view, it is necessary to practice using the learning strategy many times in order to become familiar with it. In addition, it has been reported that knowing the content of the task in advance allows for more robust (model-based)

reinforcement learning based on episodic and semantic memory to obtain rewards [43]. Therefore, learners are expected to acquire procedural knowledge more efficiently by knowing the strategy, promoting motivation to use the strategy [40], experiencing the value of using the strategy [18], and then repeatedly practicing the use of the strategy, rather than learning it blindly.

Important suggestions have been made throughout this study, but the research also had some limitations. First, this study was analyzed in a model with the use of learning strategies as the dependent variable, which suggests that the use of learning strategies and its perceived cost are originally formed through multiple learning experiences. However, the present research was conducted in order to simplify the model of within-person variance and covariance. Determining whether the subjective cognition defines the use of a learning strategy or the experience of its use shapes the subjective cognition and may be cyclical would require examination by a longitudinal study. Next, we mentioned that by focusing on the within-individual variance, it is possible to remove the errors caused by self-evaluation to some extent, such as the individual's evaluation tendency and mood at the time. On the other hand, in order to remove noise such as individual evaluation tendencies, a method that can objectively evaluate the strategies used would be desirable. Glogger, Hübner, Nückles, Renkl, and their colleagues proposed an objective measure of learning strategy using a Learning Journal [24, 44, 45]. The relationship between learning strategies and academic performance as measured by a Learning Journal tend to be the same as in previous studies of inter-individual correlations, but are unique in making it possible to assess not only the degree of strategy use but also the quality of the strategy [24]. Since it is also possible to use a Learning Journal in the within-individual variance-covariance approach proposed in this study, it is necessary to study methods, such as those used by a Learning Journal, that do not rely as much on subjective evaluation. In addition, in the within-person perspective and examination method proposed in this study, we should pay attention not only to the sample size but also to the number of items in the learning strategy, which is Level 1 of the mixed-effects model. In this regard, the sample size of this study was not as large as other learning strategy studies, and the number of items in the learning strategy was reduced due to the number of variables. The simulation results show that the method employed in this study (i.e., mixed-effects modeling) can be expected to prevent inflation in the Type I error rate [32], but the power of the test decreases with a smaller number of items [34]. Similar to the sample size of items, the sample size of participants in this study is not large enough compared to other SRL studies. The present studies used the MCMC method to address multivariate normality for estimating a large number of parameters, including variable effects with mixed-effects models. On the other hand, the adoption of MCMC methods cannot be an exemption to small samples, and of course, large sample sizes are necessary for generalization of the study results. Therefore, perceived costs and use of learning strategy suggested in this study should be interpreted with caution, including whether there are other relationships that have not been detected, and future studies should examine more learning strategy items.

Nevertheless, the present study exemplifies the interpretation and importance of intra-individual variance and covariance in learning strategies research. We hope that this study will contribute to the further development of many studies and frameworks, including the study of learning strategies.

## Acknowledgments

The survey for this study was conducted while I was a student at Hosei University. This paper is a reanalysis and reorganization of a poster presented at the 6th Biennial Meeting of the

EARLI Special Interest Group 16. I would like to thank Professor Dr. Tetsuya Fujita. I would also like to thank the teachers who cooperated in the survey, and my colleagues for their help in submitting my paper.

## Author Contributions

**Conceptualization:** Tsuyoshi Yamaguchi.

**Data curation:** Tsuyoshi Yamaguchi.

**Formal analysis:** Tsuyoshi Yamaguchi.

**Funding acquisition:** Tsuyoshi Yamaguchi.

**Investigation:** Tsuyoshi Yamaguchi.

**Methodology:** Tsuyoshi Yamaguchi.

**Writing – original draft:** Tsuyoshi Yamaguchi.

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
