## [Decision Letter · Decision Letter 0]

28 Jun 2021

PONE-D-21-02885

A paradigm shift in learning strategy research: Illustration and example of a within-person examination

PLOS ONE

Dear Dr. Yamaguchi,

Thank you for submitting your manuscript to PLOS ONE. After careful consideration, we feel that it has merit but does not fully meet PLOS ONE’s publication criteria as it currently stands. Therefore, we invite you to submit a revised version of the manuscript that addresses the points raised during the review process.

We look forward to receiving your revised manuscript.

Kind regards,

Veronica Yan, Ph.D.

Academic Editor

PLOS ONE

Additional Editor Comments:

I have now received reviews from two experts and read through the paper myself and my decision is to invite a major revision.

The consensus is that there is much to like in the basic research question of within-person examination of learning strategy use. However, there are also some concerns regarding the underlying theoretical model of strategy use (and how it may or may not fit with existing models of self-regulated learning). Reviewer 2 in particular helpfully provides a lot of useful references that you should look to integrate. There are methods-related questions that we all shared—the survey items were not reported, and I could not find them in the OSF project link either. This omission makes it very hard to judge the study design. There were also methods used that should be explained more thoroughly to help the readers better understand how to interpret results.

Each reviewer has raised issues that will require a substantial rewriting, possibly reframing, of the paper. I would also encourage you to have the paper proofread by a third party who is fluent in English and will be helpful in pointing out areas that more novice readers will require more detailed explanation. Understand that the opportunity to submit a revision does not guarantee eventual acceptance of the paper, however, and that a successful revision will depend on the degree to which your next draft thoroughly and convincingly addresses the comments from the reviews. I cannot decide on the eventual suitability of the manuscript for publication until I see the revision and how it addresses the concerns raised, so I encourage you to be sure that your next draft addresses these concerns as best you can. I will plan to send your revised manuscript to the same reviewers again.

Journal Requirements:

Reviewers' comments:

Reviewer's Responses to Questions

**Comments to the Author**

1. Is the manuscript technically sound, and do the data support the conclusions?

Reviewer #1: Yes

Reviewer #2: Partly

2. Has the statistical analysis been performed appropriately and rigorously? 

Reviewer #1: Yes

Reviewer #2: Yes

3. Have the authors made all data underlying the findings in their manuscript fully available?

Reviewer #1: Yes

Reviewer #2: Yes

4. Is the manuscript presented in an intelligible fashion and written in standard English?

Reviewer #1: No

Reviewer #2: Yes

5. Review Comments to the Author

Reviewer #1: I have great enthusiasm for the authors' arguments that within-subjects analyses need to be pursued more frequently and more rigorously than is currently the practice in many areas of psychology nd education, including the use of learning strategies. As such, I was very intrigued to review this article, in part, because I have attempted the same task of explaining the within-subjects approach to research methods to an audience that largely focuses on between-subjects methods.

I believe I understand the authors' study and the design of the data collection and analysis, but, unfortunately, it is written in a way that makes it very difficult to understand, especially for those unfamiliar with this methodology. Overall, I think a consultation with an English editor or writing assistant is in order. Although much of the highly technical parts of the paper are written precisely, several other points used English words that are not readily understandable in that context. I will try to point out some of these examples below.

A major concern I have is a lack of a clear theoretical model for learning strategy use that helps to provide a context for understanding the purpose and the meanings of the tudy and its results. The authors make good use of the literature but the theoretical grounding of this study isn't clear. I think this theoretical and practical context will help considerably to make the descriptions of the methods and the interpretation of the results more comprehensible. It seems like a largely methodological paper, but I'm not sure that is all that the authors intended.

In Lines 81-82 and again in 382-383: I fail to understand the claim that self-assessment someohow makes the effect of noice CONSTANT between strategies. I've done a good deal of research on self assessment, so this seems of great importance to me, but I fail to understand the author's claim or supporting argument. Sorry.

Line 127: what kinds of errors are the authors referring to?

Methods section, line 132 - 154: this is the section with he most confusion for me in terms of wording. Words like "noticed" or "did not notice" in relation to knowledge are not clear to me - who noticed what and how does this demonstrate knowledge? Similarly, "anyway" and "suitable" in relation to beliefs about the strategy do not make sense to me. these and the rest of this paragraph need attention.

One change that I think would make a contribution to understandability is to provide a copy of your survey (at least the English translation) so the reader can see the coding labels in the context of the questions, and, at the same time, see that the authors mean by "strategies" which are never defined or identified in the paper.

lines 173-175 are also very difficult to understand.The authors need to explain "maximal model," "exploratory" in the context of random effects hypothesis (exploratory vs what alternative?) What relationship is allowed to vary?

Referring to the learning or metacognitive strategies as "items" is appropriate for the description of the methods and data questionnaires, but it is confusing for understanding the broader strategies themselves. It is a serious limitation that the authors have not listed or described the specific strategies they include in the questionnaires and how these translate into "items".

Lines 208-210: not explaining the practical implications of Markov chains and the analysis will lose a lot of readers. The manuscript is technically sophisticated but it doesn't help communicate the results in a way that readers could readily replicate the study.

lines 356-357: Please supply a concrete example for this abstract summary.

Citation #5 looks like it is garbled in the formatting "Smarteditors" as an author name?

Reviewer #2: ##Summary##

In two survey studies, the authors examined students’ self-reported usage of various cognitive (Study 1) and metacognitive (Study 2) learning strategies. Students not only rated whether they used a particular strategy, but also how much they knew about that strategy as well as their beliefs about the costs and benefits of using that strategy. A similar pattern of results emerged for cognitive and metacognitive strategies. Using a multi-level regression approach (strategies nested within students), the authors found that there was significant variability in the degree to which a given student used each of the strategies surveyed. When a student viewed a strategy as more cumbersome, they were less likely to use that strategy. Conversely, when a student viewed a strategy as more beneficial, they were more likely to use that strategy. Finally, the more that a student knew about a strategy, the more likely they were to use that strategy.

##Evaluation##

There is a lot to like about the foundational ideas in this paper. In particular, the focus on intra-individual variability in the use of study strategies in an important direction for the self-regulated learning literature. As the authors point out, there is a growing body of research on differences between students in terms of which learning strategies they use and whether these differences in strategy usage predict differences between students in academic achievement between students (e.g., Hartwig & Dunlosky, 2012). However, I think this paper is one of the first to try to empirically model how different factors influence individual students’ study strategy choices. As such, this paper provides a useful demonstration of how to use a Bayesian multilevel modeling approach to test intra-individual research questions about self-reported study strategies.

That said, I have significant concerns 1) about the comprehensiveness of the Introduction and General Discussion and 2) about the survey items that were used. Many of my concerns could be addressed with significant writing revisions. However, I cannot determine whether my concerns with the survey items and resulting conclusions could be sufficiently addressed with writing alone unless I had more information about the survey items.

First, the Introduction and General Discussion could use a lot of further development to make it much clearer how the present studies connect to and extend the existing literature on self-regulated learning. There are several places in the manuscript where no references are provided for big claims even though such references exist. Some of the claims that need citation are listed in my ‘Other Points’ by the associated page and line number. More importantly, it wasn’t clear to me how the present study fits into the existing self-regulated learning research, particularly research on students’ study strategies. For example, there is a growing body of survey (e.g., Geller et al., 2018; Morehead et al., 2016) and observational (e.g., Karpicke, 2009; Yan et al., 2016) research examining whether, how, and to some degree why they use a range of study strategies. The present manuscript could be significantly strengthened by referencing this work and clarifying how the present studies represent an advance (e.g. pg. 12 line 351). One example of the intra-individual approach that the present manuscript advocates for comes from Blaisman, Dunlosky, and Rawson (2017) who surveyed students over the course of a semester and found that although students intend to space their studying at the start of the semester, they end up cramming a day or two before the test. The present studies could be connected to relevant survey and observational study strategy research. Was the advance of the present manuscript the study strategies surveyed, the focus on perceived cost and benefit as potential moderators study strategy use, the analytic approach, or some combination of these?

I personally was quite intrigued by the paper’s perspective of framing study decisions as weighing the relative costs and benefits of each study strategy. It is a less common way of framing study decisions and so I think it makes a nice addition to the literature. However, the impact of the paper would be stronger if the connection between the cost/benefit framework and previous self-regulated research was clarified. How does this perspective fit within existing self-regulated learning frameworks (e.g., for some reviews, see Kornell & Finn, 2016; Panadero, 2017)? For example, a leading concept in the self-regulated learning literature is that students select study strategies that feel subjectively easier rather than strategies that feel more difficult (e.g., for an empirical demonstration, see Kirk-Johnson et al., 2019). Would subjective ease of learning be considered a benefit in the cost/benefit framework?

On a related note, it was unclear to me how knowledge of each strategy was assessed in Studies 1 or 2. What was the reasoning for examining whether knowledge of a strategy predicts usage? This is in fact a ‘hot topic’ in current self-regulated learning research. For example, Yan and colleagues (2016) have a delightful paper examining whether teaching students about the value of interleaved practice changes their self-regulated study behaviors. Similarly, there has been some recent work examining whether training students about a range of effective learning strategies increases the likelihood that they use these strategies (e.g., McDaniel & Einstein, 2020). Prior research seems to suggest that the relationship between knowledge and usage is modest at best. Given that the results demonstrated a positive association between knowledge and usage of a strategy, I would highly recommend expanding the Intro and/or GD with some discussion of prior research on this association and why the results of this study may differ slightly from previous research.

My second major concern is about the survey questions that were used. Could you provide the entire cognitive and metacognitive surveys in the appendix or at least an excerpt from them? It would also be helpful to include the response options to each survey. Although some examples were given in the text, the phrasing made it quite difficult to follow. For example, what does the “belief about strategy use (anyway, suitable)” mean (p. 4 line 139)? Similarly, what does “noticed” refer to and how was it measured? (p. 5 line 145).

I was left wondering, what were the specific strategies surveyed in each study. What, specifically, were the cost, benefit, and knowledge questions? What was the theoretical motivation behind them? For example, Study 1 frames the strategies in terms of levels of processing (p. 4 line 51). How do these study strategy survey questions reflect modern work on the efficacy of various learning strategies such as elaboration, repetition, highlighting, retrieval practice, self-explanation, diagrams, etc. (e.g. Dunlosky et al., 2013). Similarly, what were the metacognitive strategies and how were they developed? If the focus of the paper wasn’t on the particular study strategies themselves, it may help to clarify this in the Introduction? Finally, how were the specific cost, benefit, and knowledge questions developed?

With these questions in mind, I could not assess the validity of these items for assessing learning strategies or the reasoning behind the use of various learning strategies. Without knowing the details of the questions, I don’t know what to conclude from the data in terms of the influences on students’ study strategy choices. As a result, without more information, I cannot assess the impact that these results would have on the self-regulated learning literature. I must note that I have some reservations. One of the items that was given in the manuscript was, “I understand the relationship between various terms.” How are students meant to answer this question? In reference to how they study in the educational psychology class that you surveyed them in? In reference to how they study for all their courses. This question could also be answered in terms of whether the student uses strategies to understand the relationship between various terms, but it could also be answered in terms of whether they think that they currently understand the relationships in their courses and not how they study. in the current class. Similarly, the question “I prefer to memorize rather than think about why” could be interpreted about what they prefer but not what they actually do.

One final set of questions—I noticed that there are relatively high correlations between the different cost measures in both Study 1 and Study 2. Multicollinearity can affect regression coefficients. This made me wonder about the interpretation of the fact that the credible intervals included 0 for Costs SS, LA, and LS. Does the data suggest that these 3 types of cost don’t influence strategy usage? Or, might the small coefficients for these 3 types of cost be caused, at least in part, by their relatively high correlation with Cost SA? Examining a variance inflation factor or some other measure of collinearity might help differentiate between these two possibilities. Relatedly, please justify whether you had sufficient level 1 and level 2 units for a full random effects model.

In short, I was quite interested in the general thrust of the research with its focus on explaining when and why a student would use one strategy over another. I hope that my suggestions help the authors communicate the impact of their research by clarifying the methods and enriching the Intro and General Discussion.

##Other Points##

• Clarify some sentences in the Abstract that were difficult to follow.

o “In cognitive strategy perspective, the results of the analysis showed that the learners perceived higher cost and avoid using the strategy which is characterized as continuing to use for the next test.”

o “Moreover, in metacognitive strategy perspective, the strategy that you have to use different strategies properly for the next test recognized more costly.”

• P.1 line 3: Please provide citations about the claim that “elaboration and metacognitive strategies can positively predict academic performance” (e.g., for a review of empirically supported study strategies, see Dunlosky et al., 2013).

• P. 1 line 8: Please further explain and provide citations for the claim that “learning strategies have 7 become a key component of research frameworks for shaping self-regulated learning.” In addition to the single citation currently provided, there has been some lovely more recent on self-regulated learning frameworks (for a review of leading self-regulated learning frameworks, see Panadero, 2017).

• P. 2 lines 10-16: I like the effort to explicitly differentiate the types of conclusions that can be drawn when students self-reported study strategies are analyzed on a between versus a within-subjects basis. Unfortunately, I found the sample interpretations provided to be very difficult to follow because they are so abstract. For example, what does “using y in strategy k mean”? Aren’t you just asking whether students use a particular strategy? Concrete examples, particularly ones taken from existing research, could greatly improve the clarity of this section of the manuscript.

• P. 2 line 17: Please include citations for the claim that “self-regulated learners naturally employ multiple learning strategies.”

• P. 2 line 31: Consider clarifying the explanation “but did not examine within-individual variance because they set individual differences to hierarchal level 1” for a less technically savvy reader.

• P. 2 line 32-36: Considering clarifying the limitations of the Obergriesser and Glogger work. The discussion about the level of analysis feels quite abstract. What specifically did the authors conclude and what could they not conclude because they used an inter-individual rather than an intra-individual analytic approach?

• P. 2 line 39: Please explain the statement “the tendency to use multiple strategies in a clustering fashion.” What does in a clustering fashion mean?

• P. 3 lines 47-57: Consider adding more details about the Murayama and Yamaguchi studies given that they most closely align with the reported studies. It may help to give concrete examples of their materials and results. For example, what does “when to use” and “how to use” mean? How were they measured? Considering your claims about the value of intra-individual research on study strategies, what are some of the conclusions that emerged from this work. Explain how these conclusions could not have been made if the analysis had been focused on inter-individual differences.

• P. 3 line 61 (and p. 12 line 263): There is a brief reference to the idea to the idea that students do not necessarily use optimal study strategies. I would recommend supporting this point with more citations, perhaps a relevant recent review. Perhaps it would be helpful to reference a concrete example or two that pertain to the study strategies surveyed in Study 1.

• p. 3 line 68: “Therefore, the present research addresses the perceived cost of using a learning strategy in examining the within-person variance-covariance.” Consider adding a sentence explaining the key question in non-statistical terms.

• P. 3 line 70: Do you mean “between-individual variance” rather than “within-individual variance”?

• P. 3 line 80: What does “evaluation characteristics” mean?

• P. 3: Consider adopting some more precise measurement language in your critique of prior survey research. It sounds like you are discussing an issue of reliability and validity.

• P. 3 line 81: I don’t agree with this claim: “However, in self-assessments, the effects of noise on the variance between strategies would be constant.” Different sources of noise can affect each item for a participant. For example, if a student just borrowed someone’s highlighter before participating in the study, they may overestimate how much they highlight in general relative to the other study strategies that they’re asked about.

• p. 4 line 127: Please clarify how many people you excluded and why. Please give concrete examples. What constitutes an error? What does “in all items of one aspect” mean? What does “knowledge regard strategy” refer to? Please justify your more subjective exclusion choices beyond excluding a participant who has too much missing data or put the same answer to every question. (I have the same feedback for Study 2).

• P. 7 line 215: “Considering the hierarchical structure that the strategy items across the participants, 20 data sets were created.” What do you mean? Did you impute missing data 20 times? How were the results then “integrated”?

• P. 8 line 241: “Note that excluding individual differences” should perhaps read “Note that except for individual differences” (see also p. 10 line 314).

• P. 8 line 255: Please clarify the phrase “although the conditions were different.” What are the conditions? Different in what way?

• P. 8 line 262: You may want to briefly explain monitoring and control and why this distinction is important.

• P. 11 line 323: Should it read “metacognitive strategies” instead of learning strategies?

• P. 11 line 325: “Perceived cost, which differs in conditional knowledge from cognitive strategies.” Is this a claim that you’re supporting with the data from your surveys? If so, how? Or, is this distinction supported by prior research?

• P. 11 line 327: “Although the conditions of time perspective, such as learning for the next test, were the same as those of the cognitive strategies, the conditions of usage were different, such as appropriate use of the strategy.” Do you mean that the pattern of coefficients were different? Please clarify.

• P. 12 line 335: Should “the why” be “the way”?

• P. 12 line 349: “Our two studies examined the within-individual variance of cognitive and metacognitive strategies and showed differences in the trends of within-individual variance for each strategy.” Consider adding a sentence to clarify what this means in behaviorally for a less statistically savvy reader.

• P. 12 line 367: Please include citations for these sentences: “In the case of inter-individual variance, the main focus was often on the quality and quantity of an individual’s motivation and their perception of the task. Traditional between-individual approaches, which attempt to identify the characteristics of individuals who use a strategy frequently, are, of course, important.”

6. PLOS authors have the option to publish the peer review history of their article (what does this mean?). If published, this will include your full peer review and any attached files.

Reviewer #1: No

Reviewer #2: No

---

## [Author Response · Author response to Decision Letter 0]

4 Oct 2021

August 30, 2021

Dr. Veronica Yan

Academic Editor

PLOS ONE

Dear Dr. Yan,

First, I apologize for the delay in submitting the revised manuscript due to my ill health. Thank you for extending my deadline, because of which I was able to complete the revised manuscript.

I am honored that you have reviewed my manuscript. This is the first paper I have written in English, and I apologize for the difficulty in reading some parts of it. As you suggested, I have revised the manuscript based on the remarks and suggestions of the two reviewers. I would like to thank you again for your cooperation.

Thank you for your consideration. I look forward to hearing from you.

Sincerely,

Tsuyoshi Yamaguchi, Ph.D.

Nippon Institute of Technology

4-1 Gakuendai, Miyashiro-machi, Minamisaitama-gun, Saitama Pref. 345-8501, Japan

+81480-33-7674 (Phone)

+81480-33-7610 (FAX)

yamaguchi.tsuyoshi@nit.ac.jp

 

In response to Reviewer #1

Thank you for your comments on my manuscript. I have learned that there are many things that I have failed to do, especially in the methods section. I have incorporated your comments and suggestions in my manuscript as follows. I would appreciate it if you could check them. I also apologize for the delay in revising this article.

A major concern I have is a lack of a clear theoretical model for learning strategy use that helps to provide a context for understanding the purpose and the meanings of the study and its results. The authors make good use of the literature but the theoretical grounding of this study isn't clear. I think this theoretical and practical context will help considerably to make the descriptions of the methods and the interpretation of the results more comprehensible. It seems like a largely methodological paper, but I'm not sure that is all that the authors intended.

 The main purpose of this paper is to propose a methodology for the study of learning strategies. On the other hand, I agree that more references should be made to the use of learning strategies, as you suggest, to clearly demonstrate the validity of the methodology. The newly added paragraphs also include descriptions pointed out and suggested by Reviewer #2.

BEFORE: N/A

 AFTER (Lines 43 - 118): Three paragraphs have been added to the "Introduction" section.

In Lines 81-82 and again in 382-383: I fail to understand the claim that self-assessment someohow makes the effect of noice CONSTANT between strategies. I've done a good deal of research on self assessment, so this seems of great importance to me, but I fail to understand the author's claim or supporting argument. Sorry.

 As you highlighted, my point was unclear in this sentence. I thought that the influence of individual response tendencies and mood on self-evaluation would affect the entire strategy (i.e., all evaluations of the strategy) rather than individual strategies. I have revised the text as follows.

BEFORE (Lines 81-82): 

However, in self-assessment, the effects of noise on the variance between strategies would be constant.

AFTER (Lines 212 - 219): 

However, even if the measurement is based on self-ratings, the variance between strategies (i.e., within individuals) is likely to be less influenced by a participant's evaluative tendencies or mood at the time than between individuals; this is because the individual's evaluation traits and moods are not expected to bring different noise to the strategies, but to affect the evaluation behavior itself (although the most recent learning or learning behavior, such as the impression of the strategy used at that time, will be noise for the within-person as well as between-person variances).

 BEFORE (Lines 382-383):

 Next, we have shown that focusing on intra-individual variance may remove some of the errors due to self-assessment, but it is better to be able to objectively evaluate the strategies used.

 AFTER (Lines 652 - 656):

 Next, we mentioned that by focusing on the within-individual variance, it is possible to remove to some extent the errors caused by self-evaluation, such as the individual's evaluation tendency and mood at the time. On the other hand, in order to remove noise such as individual evaluation tendencies, a method that can objectively evaluate the strategies used would be desirable.

Line 127: what kinds of errors are the authors referring to?

 This was definitely unclear. I added information regarding the kinds of wrong answers.

 BEFORE (Line 127):

 Participants whose survey responses included errors on more than 20% of the measurement items (i.e., 12 of 56 answer items) and/or in all items of one aspect (e.g., knowledge regarding strategy), and those who gave the same response for every question throughout the survey were excluded from analysis.

 AFTER (Lines 262 - 272): 

 Participants were excluded from the analysis based on the following three considerations:

(a) participants whose survey responses contained more than 20\\% (i.e., 12 of 56 answer items) or more errors such as non-response to a question, response to more than one numerical value for an item, or no response on a numerical value, (b) participants who were consistently non-responsive to one/more variable(s) in all eight items of the learning strategy (e.g., one of the excluded participants did not respond to all eight learning strategies, only to the question knowledge regarding strategy: $Knowledge$ "whether you '[A] were aware before' or '[B] were not aware until hearing about it on this questionnaire' of the existence of the method being asked about"), and (c) participants who gave the same response to all questions during the survey.

Methods section, line 132 - 154: this is the section with he most confusion for me in terms of wording. Words like "noticed" or "did not notice" in relation to knowledge are not clear to me - who noticed what and how does this demonstrate knowledge? Similarly, "anyway" and "suitable" in relation to beliefs about the strategy do not make sense to me. these and the rest of this paragraph need attention.

One change that I think would make a contribution to understandability is to provide a copy of your survey (at least the English translation) so the reader can see the coding labels in the context of the questions, and, at the same time, see that the authors mean by "strategies" which are never defined or identified in the paper.

 The expression in the actual questionnaire and its English translation were difficult to understand. As you suggested, we have added the English translations of the instructions and items.

 BEFORE: N/A

 AFTER: New section " Supporting information" added.

lines 173-175 are also very difficult to understand. The authors need to explain "maximal model," "exploratory" in the context of random effects hypothesis (exploratory vs what alternative?) What relationship is allowed to vary?

 There was a lack of explanation for random effects. The following explanation has been added.

 BEFORE (Lines 173-175): 

 The relationship was allowed to vary among the seven strategy aspects and each participant (i.e., random intercept and random slope model), as the maximal model has been found appropriate if the hypothesis regarding random effects was exploratory [19].

 AFTER (Lines 318 - ): 

 There are some noteworthy points when building a model that takes random effects into account. Whether it is a random-participant or random-item effect, it is necessary to establish whether the differences in participants or items affect the scores of the use of the learning strategy (i.e., the random intercept), the relationship between the explanatory and objective variables (i.e., the random slope), or both. In addition, it is possible to separately set random effects for the participants and items, such as only the intercept for the random-participant effect, or both the intercept and the slope for the random-item effect. As mentioned earlier, this study does not have a rationale for setting random effects because it examines the within-individual variance (between-item variance) in learning strategy research. Thus, because of the exploratory stance of this study toward random effects, we decided to use the maximal model with random effects of intercept and slope in both the random-participant and random-item effects [34].

Referring to the learning or metacognitive strategies as "items" is appropriate for the description of the methods and data questionnaires, but it is confusing for understanding the broader strategies themselves. It is a serious limitation that the authors have not listed or described the specific strategies they include in the questionnaires and how these translate into "items".

 As you pointed out, there was a great deal of ambiguity between the learning strategies discussed in the paper and the items that were actually measured. I presented a list of items measured in the "Supporting information" section in response to your previous comment. I am not sure if this will resolve the reviewers' concerns, and I may not have understood your concern well enough.

Lines 208-210: not explaining the practical implications of Markov chains and the analysis will lose a lot of readers. The manuscript is technically sophisticated but it doesn't help communicate the results in a way that readers could readily replicate the study.

 Following your suggestion, I have added a basic explanation of MCMC and the setting of this study.

BEFORE: N/A

 AFTER (Lines 364 - 373): 

 The MCMC method is an algorithm for sampling probability distributions using random numbers, and it can be used instead of the maximum likelihood estimation method to obtain estimates and their intervals. The estimates presented in this study report the median of the posterior distribution. In addition, this study presents a credible interval, which is the range in which the population is included with 95\\% probability, as an alternative to the confidence interval required by the maximum likelihood estimation method; additionally, it refers to the relationship between the explanatory and objective variables based on whether or not the credible interval contains zero. The calculation involved a minimum of 10,000 iterations, and the first half of each chain was discarded as part of the burn-in phase in Mplus ver. 8.3 [36]. The MCMC algorithm was repeated four times with different initial values. The seed value was set to 1015.

lines 356-357: Please supply a concrete example for this abstract summary.

 I added a sentence with a specific example right after the sentence you pointed out.

BEFORE: N/A

 AFTER (Lines 556 - 559): 

 For example, between-person correlations are interpreted as "the more troublesome the strategy is, the less likely it is to be used," while within-person correlations are interpreted as "the more troublesome a strategy is, the less likely it is to be used compared to other strategies that are not troublesome."

Citation #5 looks like it is garbled in the formatting "Smarteditors" as an author name?

 This paper is written in LATEX and submitted to the system in PDF format. I checked the submitted manuscript, and it was displayed correctly. I am sorry that I was unable to resolve this problem.

 

In response to Reviewer #2

I am grateful for the various references to improve my manuscript and to assist in developing the discussion. I was able to add not only the references you mentioned, but also many more. I think this helped me to make a stronger case for the important points of my paper. I also tried to reflect the other points you raised as much as I could. I would like to ask you to check them. I also apologize for the delay in revising the paper. 

## First major concern ##

First, the Introduction and General Discussion could use a lot of further development to make it much clearer how the present studies connect to and extend the existing literature on self-regulated learning. ･･･ Was the advance of the present manuscript the study strategies surveyed, the focus on perceived cost and benefit as potential moderators study strategy use, the analytic approach, or some combination of these?

I personally was quite intrigued by the paper’s perspective of framing study decisions as weighing the relative costs and benefits of each study strategy. ･･･ Would subjective ease of learning be considered a benefit in the cost/benefit framework?

On a related note, it was unclear to me how knowledge of each strategy was assessed in Studies 1 or 2. ･･･ Given that the results demonstrated a positive association between knowledge and usage of a strategy, I would highly recommend expanding the Intro and/or GD with some discussion of prior research on this association and why the results of this study may differ slightly from previous research.

 The main purpose of this study was not to compare the relative frequency of use of learning strategies among individuals, but to determine how strategies are used within an individual, and to argue for the importance of a within-person approach for the main purpose. However, you are right; learning strategies are now inseparable from the framework of self-regulated learning. Therefore, as you suggested, I have added a new argument based on the theoretical relationship with self-regulated learning, the contribution of this study, and the empirical studies you mentioned. I have added three new paragraphs in the "Introduction" and two new paragraphs in the "General Discussion."

BEFORE : N/A

 AFTER: Some new paragraphs have been added to the “Introduction (lines 43 - 118)” and “General Discussion (lines 583 - 643).”

## Second major concern ##

My second major concern is about the survey questions that were used. ･･･ Similarly, what does “noticed” refer to and how was it measured? (p. 5 line 145).

I was left wondering, what were the specific strategies surveyed in each study. ･･･ Finally, how were the specific cost, benefit, and knowledge questions developed?

With these questions in mind, I could not assess the validity of these items for assessing learning strategies or the reasoning behind the use of various learning strategies. ･･･ Similarly, the question “I prefer to memorize rather than think about why” could be interpreted about what they prefer but not what they actually do.

 First, a new section, "Supporting Information," was added to include the actual teaching statements and questions that the participants saw. As the questionnaire survey was conducted in Japanese, we hired an English proofreading company to perform back translation. However, unfortunately, it does not match the strategies that were covered in the literature you referred to. The cognitive strategies cover understanding the meaning of the learning content or memorizing the content text, and the metacognitive strategies cover monitoring and control. I would like to ask you to confirm whether you agree with our findings.

BEFORE: N/A

 AFTER: A new section "Supporting Information" has been added between "General Discussion" and "Acknowledgments".

 Through your remarks, I realized that it was also problematic that there was no specific reference to the strategies to be measured, as the effects of specific learning strategies and learners' beliefs were not the subject of this study. Therefore, at the end of the first paragraph of the original "Introduction," I provided a brief description of the strategies used in the measurement, and further stated that the interest of this study is intra-individual variance of learning strategies, and that we did not intend to examine or refer to any particular learning strategy.

BEFORE: N/A

 AFTER (Lines 27 - 42): 

 Thus, the purpose of this study was to provide a within-person perspective on the use of learning strategies, such as why a learner used a particular strategy from among multiple strategies.

Therefore, the present research did not focus on the effects of specific strategies or individual differences among learners, and was not concerned with the type of strategies used, as it is desirable that there be variance in the strategies used within individuals. In this study, two surveys were conducted, each measuring cognitive and metacognitive strategies. The cognitive strategies focused on the levels of processing of memory, and includes a deep processing strategy focusing on understanding the meaning of information in the content to be learned, and a shallow processing strategy focusing on memorizing the text of the content [7]. The metacognitive strategies consisted of monitoring, in which the learner objectively grasps and evaluates their learning behavior, and control, in which the learner changes and maintains their learning behavior and sets behavioral guidelines [8]. Although the strategies may vary from those measured in empirical studies on SRL [9 – 12], this study is one of the few that provide a perspective and way to examine the within-person (between-strategy) variance and covariance of learning strategies, based on which the items were measured.

 Unfortunately, as disclosed in the "Supporting Information" section, the instructional text is not designed to be subject-specific for learners. This was a required response in terms of privacy protection. Although this is only an excuse, I asked the class instructor to allow me to implement the lesson during class time so that the learners' expected learning content would be in the subject that I implemented. I hope you will understand that this was the best I could do. In addition, the survey items were not created from scratch by Yamaguchi (2016), but were created by referring to some previous studies in Japanese on levels of processing and metacognitive activities. As I was looking at the items of the learning strategies in Japanese, it did not feel strange to me, but now I agree with your point that the items are not clear as learning behaviors and learners may have trouble answering them. On the other hand, it is possible that this is my fault in translating Japanese into English.

BEFORE: 

 (Lines 134-137) Four items were strategies for deep information processing (e.g., “I understand the relationship between various terms”), and the other four items were strategies for shallow information processing (e.g., “I prefer to memorize rather than think about why”) [16].

 (Lines 284-287) Seven items were measured: four items for strategies for monitoring one’s own learning activities (e.g., “I try to figure out how much I understand”), and other three items about strategies for controlling one’s own learning behavior (e.g., “I try to use different study methods for each task.”).

 AFTER: 

 (Lines 278 - 281) Four items were strategies for deep information processing (e.g., “Memorize the process and big picture of what you learned”), and the other four items were strategies for shallow information processing (e.g., “Prioritize memorization over thinking about why it is that way”) [31].

 (Lines 467 - 471) Seven items were measured: four items for strategies for monitoring one’s own learning activities (e.g., “Predict how much you will probably be able to accomplish”), and other three items about strategies for controlling one’s own learning behavior (e.g., “Try to use an appropriate study method for each topic”).

## Third major concern ##

One final set of questions—I noticed that there are relatively high correlations between the different cost measures in both Study 1 and Study 2. ･･･ Relatedly, please justify whether you had sufficient level 1 and level 2 units for a full random effects model.

 First, the variance inflation factor (VIF) was calculated from the correlation coefficients shown in Tables 2 and 5. As a result, the concern about multicollinearity seems to have been dispelled. Secondly, due to my lack of study, I am not sure if I have responded appropriately to your request "please justify whether you had sufficient level 1 and level 2 units for a full random effects model." As with the sample size, the number of items in Level 1 was small, and I mentioned the problems that arose there in the General Discussion.

BEFORE: N/A

 AFTER: 

 (Lines 411 - 413) When the variance inflation factor (VIF) between explanatory variables was calculated from the within-person correlation coefficients, the highest correlation coefficient was .81, with a VIF of 2.91, indicating between CostLA and CostLS.

 (Lines 501 - 504) When the VIF between explanatory variables was calculated from the within-person correlation coefficients, the highest correlation coefficient was .75, with a VIF of 2.25, indicating between CostLA and CostLS (i.e., no concern for multicollinearity in Study 2, as in Study 1).

 (Lines 664 - 675) In addition, in the within-person perspective and examination method proposed in this study, we should pay attention not only to the amount of sample size but also to the number of items in the learning strategy, which is Level 1 of the mixed-effects model. In this regard, the sample size of this study was not as large as other learning strategy studies, and the number of items in the learning strategy was reduced due to the number of variables. The simulation results show that the method employed in this study (i.e., mixed-effects modeling) can be expected to prevent inflation in the Type I error rate [32], but the power of the test decreases with a smaller number of items [34]. Therefore, perceived costs and use of learning strategy suggested in this study should be interpreted with caution, including whether there are other relationships that have not been detected, and future studies should examine more learning strategy items.

##Other Points##

• Clarify some sentences in the Abstract that were difficult to follow.

o “In cognitive strategy perspective, the results of the analysis showed that the learners perceived higher cost and avoid using the strategy which is characterized as continuing to use for the next test.”

o “Moreover, in metacognitive strategy perspective, the strategy that you have to use different strategies properly for the next test recognized more costly.”

 I rewrote this as follows:

The analysis revealed that in terms of cognitive strategies (Study 1), learners avoided using learning strategies that they perceived as high cost to "use for the next test" and "keep using the strategy anyway, regardless of the content".

On the other hand, in metacognitive strategies (Study 2), students avoided using learning strategies that they perceived as costly to "use for the next test" and "use as appropriate for the content".

• P.1 line 3: Please provide citations about the claim that “elaboration and metacognitive strategies can positively predict academic performance” (e.g., for a review of empirically supported study strategies, see Dunlosky et al., 2013).

 The following references, which have already been cited, have been added to the cited sources.

Pintrich PR, De Groot EV. Motivational and self-regulated learning components of classroom academic performance. J Educ Psychol. 1990 Mar;82(1):33–40.

• P. 1 line 8: Please further explain and provide citations for the claim that “learning strategies have 7 become a key component of research frameworks for shaping self-regulated learning.” In addition to the single citation currently provided, there has been some lovely more recent on self-regulated learning frameworks (for a review of leading self-regulated learning frameworks, see Panadero, 2017).

 After the relevant sentence, I inserted a sentence to explain the logic. In addition, as the references cited here were also cited previously, I have removed the citation here (i.e., Pintrich & De Groot, 1990) and focused on the review papers you referred to.

(Lines 8 - 11) Recent reviews have proposed several theoretical models for SRL, all of which assume that learning strategies play some role [2], and some theories explicitly incorporate specific learning strategies into their models [3 – 5].

• P. 2 lines 10-16: I like the effort to explicitly differentiate the types of conclusions that can be drawn when students self-reported study strategies are analyzed on a between versus a within-subjects basis. Unfortunately, I found the sample interpretations provided to be very difficult to follow because they are so abstract. For example, what does “using y in strategy k mean”? Aren’t you just asking whether students use a particular strategy? Concrete examples, particularly ones taken from existing research, could greatly improve the clarity of this section of the manuscript.

 I was trying to generalize, but as you pointed out, I ended up with an incomprehensible expression. I have added specific example sentences for both inter-personal and intra-personal. After the explanations in the notes, I have added specific examples based on the tutorial by Murayama (2017) as examples.

(Lines 13 - 16) "･･･e.g., Learners with a high perceived cost of a learning strategy [I focus on memorizing the facts without considering why] tend not to use that strategy as much as learners with a low cost perception [6]" (see Fig 1).

(Lines 19 - 24) "･･･(e.g., If the perceived cost of learning strategy [When I learn a new historical event, I try to understand what actually happened] is higher than the perceived cost of learning strategy [I focus on memorizing the facts without considering why] within a single learner, the more costly learning strategy will not be used [6]),” may be essential to promoting the use of learning strategies (see Fig 2).

• P. 2 line 17: Please include citations for the claim that “self-regulated learners naturally employ multiple learning strategies.”

 The following references have been added.

Boekaerts M. Self-regulated learning at the junction of cognition and motivation. Eur Psychol. 1996 Jun;2(2):100–112.

• P. 2 line 31: Consider clarifying the explanation “but did not examine within-individual variance because they set individual differences to hierarchal level 1” for a less technically savvy reader.

 After the original description, I added a specific expression that ties in with the data.

(Lines 131 - 136) If we do not assume hierarchy in our data, as it is traditionally done, and consider the correlation between two variables with one row of the dataset as a single participant, then we are considering the variance between individuals. Similarly, if a dataset contains one participant's data in one row, even if there is hierarchy (higher levels), we cannot examine the variance within individuals, because even the smallest unit can only represent the individual differences.

• P. 2 line 32-36: Considering clarifying the limitations of the Obergriesser and Glogger work. The discussion about the level of analysis feels quite abstract. What specifically did the authors conclude and what could they not conclude because they used an inter-individual rather than an intra-individual analytic approach?

 The following corrections and additions have been made to the relevant sections. The two studies mentioned address the variance within individuals of learning strategies, but they differ from this study in that they are not multiple strategies; that is, they do not examine the variance among strategies within an individual.

(Lines 137 - 143) Hence, research on learning strategies has begun to address variation and variance within individuals while recognizing the importance of exploring intra-individual strategies. In particular, the contribution of these studies is to reveal the change in the same strategy over time and the variance in each content of the same strategy. On the other hand, it would be difficult to give any indication of the variance that can occur within an individual between multiple strategies at the same time and in the same situation, that is, within-person perspectives of learning strategy use.

• P. 2 line 39: Please explain the statement “the tendency to use multiple strategies in a clustering fashion.” What does in a clustering fashion mean?

 It was indeed a difficult expression to understand. I have corrected it by adding the following text to the relevant part.

(Lines 145 - 149) Although Glogger and colleagues did indeed measure multiple strategies, they performed a cluster analysis using the degree of use of those strategies to identify distinctive clusters of learners who used each strategy with similar frequency [24]; this approach only reveals group characteristics and does not examine the strategy variance of particular individuals.

• P. 3 lines 47-57: Consider adding more details about the Murayama and Yamaguchi studies given that they most closely align with the reported studies. It may help to give concrete examples of their materials and results. For example, what does “when to use” and “how to use” mean? How were they measured? Considering your claims about the value of intra-individual research on study strategies, what are some of the conclusions that emerged from this work. Explain how these conclusions could not have been made if the analysis had been focused on inter-individual differences.

 I followed your suggestion and added more detailed information.

(Lines 160 - 182) Murayama and colleagues argued for the need for an intra-individual approach in educational psychology: As an example, they mentioned inter- and intra-personal approaches to the use of learning strategies [6]. Their paper discussed data from a questionnaire survey measuring the perceived benefit, perceived cost, and usage of a learning strategy on 17 different learning strategy items using a Likert scale [25]. Conditional knowledge, such as "when is the learning directed at," was taken into account in the perceived benefit, and the inter- and intra-individual approaches were described with the use of the learning strategy as the objective variable and the subjective perceptions of the three learning strategies as explanatory variables, such as "is the strategy effective for learning for the next test (short-term utility)," "is the strategy effective for future learning (long-term utility)," and "is the strategy hard to use (cost)." Depending on the research question, such as "promoting the use of learning strategies," we can address the question more directly, such as what strategies are used most often, with a within-person approach to "compare across strategies," rather than with a between-person approach to "compare across learners" to learn who uses a particular strategy most frequently. Yamaguchi used a questionnaire survey to measure 15 items (8 cognitive strategies + 7 metacognitive strategies) of various learning strategies for obtaining an interpretation of the within-person approach presented by Murayama and colleagues, and proposed a 2 × 2 perceived benefit by adding conditional knowledge of usage inspired by the SRL framework, such as "how to use it effectively (use it anyway without considering the learning content OR use it according to the learning content)," to the temporal conditional knowledge of "when is it effective (short term [next test] OR long term [future])" that they showed [26].

• P. 3 line 61 (and p. 12 line 263): There is a brief reference to the idea to the idea that students do not necessarily use optimal study strategies. I would recommend supporting this point with more citations, perhaps a relevant recent review. Perhaps it would be helpful to reference a concrete example or two that pertain to the study strategies surveyed in Study 1.

 I was able to add the references you mentioned to the relevant sections. Additionally, unfortunately, I could not determine which part "p. 12 line 263" refers to.

(Lines 569 - 577) To date, many previous studies have shown that learners may not always perceive theoretically effective learning strategies to be effective [9 – 12, 18 - 20]. (In this study, the deep processing strategy in Study 1 [e.g., "Understand the relationships between various terms"] and the metacognitive strategy in Study 2 in general were theoretically effective strategies. As shown in Table 1 and 4, unlike previous studies, participants in this study seemed to perceive theoretically effective strategies as subjectively effective as well.) Fortunately, however, there is some evidence that approaching learners about their misperceptions can promote the use of theoretically effective strategies [18].

• p. 3 line 68: “Therefore, the present research addresses the perceived cost of using a learning strategy in examining the within-person variance-covariance.” Consider adding a sentence explaining the key question in non-statistical terms.

 New explanatory text has been added to the relevant sections.

(Lines 196 - 198) (i.e., we examine variations in the degree of use and subjective cost among multiple strategies within a learner, as well as the within-person relationship between use of learning strategy and perceived cost)

• P. 3 line 70: Do you mean “between-individual variance” rather than “within-individual variance”?

 I apologize for the very confusing wording. The relevant part has been corrected as follows. I only tried to describe the advantages of the within-person approach addressed in this study.

(Lines 199 - 201) The framework addressed in the present research in assessments of learning strategy use, which typically focuses on within-individual variance, has some methodological advantages.

• P. 3 line 80: What does “evaluation characteristics” mean?

 I was referring to the inherent tendency of participants to answer questions with high/low answers. I added a note to that effect.

(Lines 210 - 212) The effects of noise, such as mood and evaluation characteristics while responding to a questionnaire item with a high or low score, can vary in strength between individuals.

• P. 3: Consider adopting some more precise measurement language in your critique of prior survey research. It sounds like you are discussing an issue of reliability and validity.

 I have made the following corrections, and I hope I have adequately addressed your points.

(Lines 201 - 204) One method to determine the learning strategies used by learners is utilizing a questionnaire approach. In general, such self-assessments include noise such as individual response tendencies and moods, and the measured responses may differ from third party impressions or evaluations based on objective measurements.

• P. 3 line 81: I don’t agree with this claim: “However, in self-assessments, the effects of noise on the variance between strategies would be constant.” Different sources of noise can affect each item for a participant. For example, if a student just borrowed someone’s highlighter before participating in the study, they may overestimate how much they highlight in general relative to the other study strategies that they’re asked about.

 This point was rewritten in detail by Reviewer #1. I think the description has been written taking into account your comments.

(Lines 212 - 219) However, even if the measurement is based on self-ratings, the variance between strategies (i.e., within individuals) is likely to be less influenced by a participant's evaluative tendencies or mood at the time than between individuals; this is because the individual's evaluation traits and moods are not expected to bring different noise to the strategies, but to affect the evaluation behavior itself (although the most recent learning or learning behavior, such as the impression of the strategy used at that time, will be noise for the within-person as well as between-person variances).

• p. 4 line 127: Please clarify how many people you excluded and why. Please give concrete examples. What constitutes an error? What does “in all items of one aspect” mean? What does “knowledge regard strategy” refer to? Please justify your more subjective exclusion choices beyond excluding a participant who has too much missing data or put the same answer to every question. (I have the same feedback for Study 2).

 I have answered each of the questions you raised in detail.

(Lines 262 - 275) Participants were excluded from the analysis based on the following three considerations: (a) participants whose survey responses contained more than 20% (i.e., 12 of 56 answer items) or more errors such as non-response to a question, response to more than one numerical value for an item, or no response on a numerical value, (b) participants who were consistently non-responsive to one/more variable(s) in all eight items of the learning strategy (e.g., one of the excluded participants did not respond to all eight learning strategies, only to the question knowledge regarding strategy: Knowledge "whether you '[A] were aware before' or '[B] were not aware until hearing about it on this questionnaire' of the existence of the method being asked about"), and (c) participants who gave the same response to all questions during the survey. Finally, 151 students’ answers were used for the analysis (34 women; Mage = 18.68, SDage = 1.11, Rangeage = 17–28), except for 12 students. The data for analysis included answers from 151 students regarding eight strategy items for a total of 1,208 responses to Study 1, including those with missing values.

(Lines 454 - 464) Participants were excluded from the analysis based on the following three considerations: (a) participants whose survey responses contained more than 20% (i.e., 10 of 49 answer items) or more errors such as non-response to a question item, response to more than one numerical value for an item, or no response on a numerical value, (b) participants who were consistently non-responsive to one/more variable(s) in all seven items of the learning strategy, and (c) participants who gave the same response to all questions during the survey. Finally, 158 students' answers were used for analysis (92 women; Mage = 19.25, SDage = 1.90, Rangeage = 18-38, Eight participants had unknown age), except for 30 students. The data for analysis included answers from 158 students regarding seven strategy items for a total of 1,106 responses in to study 2, including those with missing values.

• P. 7 line 215: “Considering the hierarchical structure that the strategy items across the participants, 20 data sets were created.” What do you mean? Did you impute missing data 20 times? How were the results then “integrated”?

 The description of the multiple substitution method was unclear. It has been corrected as follows.

(Lines 378 - 384) After considering the crossed data structure of the strategy items and participants, we substituted the regression prediction plus the residuals for the missing data points. This procedure was repeated 20 times to create 20 data sets with different assigned values, and the analysis was performed on each data set. The integrated results of the 20 analyses are reported in this study. Third, an analysis of the mixed-effects model was performed for each data set created by the multiple imputation method, and the results were integrated.

• P. 8 line 241: “Note that excluding individual differences” should perhaps read “Note that except for individual differences” (see also p. 10 line 314).

 Thank you for providing the revised proposal. The corrections have been reflected in the respective sections.

• P. 8 line 255: Please clarify the phrase “although the conditions were different.” What are the conditions? Different in what way?

 As you pointed out, the description was too short, so I expanded it.

(Lines 424 - 431) In other words, referring to previous studies, we also established 2 (when: short-term OR long-term) x 2 (how: all times OR fits situation) conditional knowledge about perceived cost for the use of learning strategies [6, 25, 26]; however, the between-strategy correlation coefficients among the four variables were high (to the extent that we did not have to worry about multicollinearity), and it is possible that learners do not discriminate between strategies measured in Study 1 least and conditional knowledge such as "when" and "how."

• P. 8 line 262: You may want to briefly explain monitoring and control and why this distinction is important.

 I have added a note about monitoring and control.

(Lines 439 - 444) Monitoring is an activity in which the learner reflects on and evaluates their own learning situation and past behavior, while control is an activity in which the learner reviews their own learning behavior and future plans in response to monitoring. As it is desirable for metacognitive activities to activate both of these activities, we measured both activities so that we can examine which one is not being used.

• P. 11 line 323: Should it read “metacognitive strategies” instead of learning strategies?

 I am grateful to you; your interpretation is right.

• P. 11 line 325: “Perceived cost, which differs in conditional knowledge from cognitive strategies.” Is this a claim that you’re supporting with the data from your surveys? If so, how? Or, is this distinction supported by prior research?

• P. 11 line 327: “Although the conditions of time perspective, such as learning for the next test, were the same as those of the cognitive strategies, the conditions of usage were different, such as appropriate use of the strategy.” Do you mean that the pattern of coefficients were different? Please clarify.

 In response to these two questions and requests, I have cited the relevant results of this study as follows.

(Lines 513 - 518) Although the conditions of time perspective, such as learning for the next test, were the same as those of the cognitive strategies, the conditions of usage were different, such as appropriate use of the strategy (\\gamma_{10}\\left[Cost_{SA}\\right]=-0.16 in study1, whereas \\gamma_{20}\\left[Cost_{SS}\\right]=-0.21 in study 2; since both of these did not include 0 in the 95% confidence interval, it is likely that there is a negative effect on the use of learning strategies).

• P. 12 line 335: Should “the why” be “the way”?

 I fixed the part you pointed out. I apologize for the inconvenience.

• P. 12 line 349: “Our two studies examined the within-individual variance of cognitive and metacognitive strategies and showed differences in the trends of within-individual variance for each strategy.” Consider adding a sentence to clarify what this means in behaviorally for a less statistically savvy reader.

 I added the following sentence after the one you pointed out.

(Lines 540 - 549) Specifically, the following differences were suggested between the cognitive and metacognitive strategies within a (single) learner: For cognitive strategies, the higher the subjective value of "it's hard to keep using it anyway for the next test," the less likely they were to use the strategy; On the other hand, metacognitive strategies showed the same correlation as cognitive strategies in short-term "when" conditional knowledge, such as "use it for the next test," but the higher the cost of "how" conditional knowledge, such as "use it according to the content," the less likely they were to use the strategy. Thus, for the variance among multiple strategies within a learner, the factors causing the variance differed between cognitive and metacognitive strategies.

• P. 12 line 367: Please include citations for these sentences: “In the case of inter-individual variance, the main focus was often on the quality and quantity of an individual’s motivation and their perception of the task. Traditional between-individual approaches, which attempt to identify the characteristics of individuals who use a strategy frequently, are, of course, important.”

 The following references, already cited elsewhere, have been added.

Pintrich PR, De Groot EV. Motivational and self-regulated learning componentsof classroom academic performance. J Educ Psychol. 1990 Mar;82(1):33–40.

Geller J, Toftness AR, Armstrong PI, Carpenter SK, Manz CL, Coffman CR,Lamm MH. Study strategies and beliefs about learning as a function of academic achievement and achievement goals. Memory. 2018;26(5):683–690.

---

## [Decision Letter · Decision Letter 1]

6 May 2022

PONE-D-21-02885R1A paradigm shift in learning strategy research: Illustration and example of a within-person examinationPLOS ONE

Dear Dr. Yamaguchi,

Thank you for submitting your manuscript to PLOS ONE. After careful consideration, we feel that it has merit but does not fully meet PLOS ONE’s publication criteria as it currently stands. Therefore, we invite you to submit a revised version of the manuscript that addresses the points raised during the review process.

Both reviewers have reviewed the manuscript again, and appreciate the revisions made. They have both raised additional points, which you can find below.

We look forward to receiving your revised manuscript.

Kind regards,

Hanna Landenmark

Staff Editor

PLOS ONE

Journal Requirements:

Reviewers' comments:

Reviewer's Responses to Questions

**Comments to the Author**

1. If the authors have adequately addressed your comments raised in a previous round of review and you feel that this manuscript is now acceptable for publication, you may indicate that here to bypass the “Comments to the Author” section, enter your conflict of interest statement in the “Confidential to Editor” section, and submit your "Accept" recommendation.

Reviewer #1: All comments have been addressed

Reviewer #2: (No Response)

2. Is the manuscript technically sound, and do the data support the conclusions?

Reviewer #1: Partly

Reviewer #2: Yes

3. Has the statistical analysis been performed appropriately and rigorously? 

Reviewer #1: Yes

Reviewer #2: Yes

4. Have the authors made all data underlying the findings in their manuscript fully available?

Reviewer #1: Yes

Reviewer #2: Yes

5. Is the manuscript presented in an intelligible fashion and written in standard English?

Reviewer #1: No

Reviewer #2: (No Response)

6. Review Comments to the Author

Reviewer #1: I really appreciate th within subjects approach in these studies and agree with the author that within-subjects approaches are complementary to between-subjects methods. The mathematics is challeging for many readers but appears to me to be appropriately applied.

The writing is generally good, though technical. There are a few places where I thin the translation to English has been uncertain. The most important is the use of "anyway" in labeling the study variables in contrast to "suitable". I understand the use of 'suitable' but "anyway' doesn't fit as the opposite of 'suitable.' I would recommend the author review and reconsider tis label. More generally, a proofreading for English clarity by another, independent reader would be helpful.

Technically, the only concern I have is for the adequacy of the sample size to support the computation of so many parameters and then modeling them in multiple ways. It would be helpful for the author to address the degree of over or under determination in the it of the data to the multiple parameters.

Reviewer #2: I have reviewed a version of this manuscript before and commend the authors for the significant revisions that have been made from the earlier version. I continue to appreciate the focus on the importance of examining the way context influences how students choose which study strategies and when. This point, in combination with the result that perceived difficulty of implementation had a strong negative correlation with strategy use, make this paper an interesting and important addition to the SRL literature.

The way that the author has clarified in the Intro that this is a methodological paper (promoting intra-individual analyses rather than inter-individual analyses) rather than an examination of particular learning/metacognitive strategies was particularly helpful. The author also made significant steps to position the work within broader research on self-regulated learning and decision making and such changes were largely effective. For example, the Intro includes a simple, yet productive framework (and cites previous work accordingly) suggesting that a student's study strategies are influenced by their a) knowledge that the strategy exists, b) their procedural knowledge of when and how to use that strategy, and c) if they've used the strategy, their subjective experience of whether the strategy works for them. How these three influences on strategy use align with broader theories of self-regulated learning (e.g., Zimmerman's SRL model) was also added and improves the paper. The author has also made significant improvements to the clarity of the materials, methods, analytical approach and results. For these reasons, I would recommend publication upon some more minor changes:

- Thank you for providing the Bayesian credible intervals for the L1 predictors. Am I reading this right: many of the CIs for costs of metacognitive strategies (SA, LA, SS) include 0. Does this mean that the costs of metacognitive strategies (other than SS) largely do not predict usage? If that is the correct interpretation, I would recommend emphasizing this important result more clearly in Study 2 and hypothesizing why more clearly in the discussion. The author compellingly explains why perceived costs often drive decision making. Why might the relationship between costs and use be near 0 when it comes to metacognitive strategies?

- On a similar note, although the strengths of the correlations were smaller for metacognitive strategies than cognitive strategies, I'd be hesitant to compare these correlations because different students responded to the cognitive and metacognitive surveys, I believe. The manuscript currently suggests that "Thus, for the variance among multiple strategies within a learner, the factors causing the variance differed between cognitive and metacognitive strategies." (line 547). I'd rewrite this sentence for clarity and also temper this language because I don't think that you have evidence to directly conclude that the factors that influence cognitive and metacognitive strategy use are different.

7. PLOS authors have the option to publish the peer review history of their article (what does this mean?). If published, this will include your full peer review and any attached files.

Reviewer #1: No

Reviewer #2: No

---

## [Author Response · Author response to Decision Letter 1]

6 Jun 2022

In response to Reviewer #1

Thank you for reviewing my manuscript. I agree that the mathematical model for examining intra-individual variance has become complicated, and I tried to emphasize the importance of the intra-individual approach with plenty of descriptions and figures to complement it. I am honored that you appreciated that. I have prepared a response to your points below. 

The writing is generally good, though technical. There are a few places where I think the translation to English has been uncertain. The most important is the use of "anyway" in labeling the study variables in contrast to "suitable". I understand the use of "suitable" but "anyway" doesn't fit as the opposite of "suitable." I would recommend the author review and reconsider tis label. More generally, a proofreading for English clarity by another, independent reader would be helpful.

 Thank you for pointing this out. First, I asked the proofreader to review and revise all the text in my manuscript to the extent that no changes were made to the content. Second, I have removed the expression "anyway" and replaced it with "irrelevant". With this change, the sentences about "anyway" has also been replaced with sentences about "irrelevant". Also, variables which used have been adjusted to use, so CostSA became CostSI.

BEFORE: Notation of “anyway” and the sentences before and after it

AFTER: Notation of “irrelevant” and the sentences before and after it

Technically, the only concern I have is for the adequacy of the sample size to support the computation of so many parameters and then modeling them in multiple ways. It would be helpful for the author to address the degree of over or under determination in the it of the data to the multiple parameters.

 To estimate multiple parameters, the study employed a Markov chain Monte Carlo method, which passed the problem of the normality assumption for each parameter. However, as you pointed out, we did not discuss the necessary sample size. I have added a description to address this. In mixed-effects models, in addition to the sample size of participants, the number of items/stimuli that cross within participants is relevant to parameter estimation. Therefore, we discuss the number of participants and the number of items in terms of general sampling. 

BEFORE: N/A

AFTER: page 19, line 697.

Similar to the sample size of items, the sample size of participants in this study is not large enough compared to other SRL studies. This study used MCMC methods to address multiple normality for estimating a large number of parameters, including variable effects with mixed effects models. On the other hand, the adoption of MCMC methods cannot be an exemption to small samples, and of course, large sample sizes are necessary for generalization of the study results. 

In response to Reviewer #2

Thank you again for reviewing this paper. Your important remarks and numerous suggestions have helped me to improve it. I appreciate your statement that the consistency of the model proposed in this paper with other SRL studies and the position of this paper as an SRL study is now very clear. I have revised the paper based on your comments and suggestions.

Thank you for providing the Bayesian credible intervals for the L1 predictors. Am I reading this right: many of the CIs for costs of metacognitive strategies (SA, LA, SS) include 0. Does this mean that the costs of metacognitive strategies (other than SS) largely do not predict usage? If that is the correct interpretation, I would recommend emphasizing this important result more clearly in Study 2 and hypothesizing why more clearly in the discussion. The author compellingly explains why perceived costs often drive decision making. Why might the relationship between costs and use be near 0 when it comes to metacognitive strategies?

 The presentation of the Bayesian credible intervals makes the results of this paper clearer. Thank you for your suggestion. In the metacognitive strategies, variables that included 0 in the confidence interval (i.e., other than CostSS) did not predict the degree of use for the learning strategy. As you pointed out, I did not explicitly state my interpretation of this result. I have added my interpretation below and would appreciate your review. Note that in Study 1 (Cognitive Strategies), as in Study 2, the results were not explicitly stated and there was no clear interpretation, so I have added them.

BEFORE: N/A

AFTER in Study 1: page 11, line 422.

Participants are more likely to associate cognitive strategies like memorization and comprehension with the study they are completing because these strategies are directly related to the acquisition of learning content. Therefore, a short time condition (e.g., the next test) and a condition in which participants did not consider the appropriate use of the cognitive strategy (e.g., just study for now) may have predicted the reduced use of the cognitive strategy. In other words, it is possible that the frequency of cognitive strategy use decreases only in the "short-term" and the "unconceivable situation" conditions in the subjective perceived cost held by the individual learner, but not in the other conditions.

AFTER in Study 2: page 15, line 527.

In Study 2, only CostSS predicted to Used, and the rest of CostSI, CostLI and CostLS were determined not to predict strategy use because the 95% Bayesian credible interval contained 0. Metacognitive strategies have a role in controlling learning behavior rather than being involved in learning content. As with the cognitive strategies, the distant future condition does not predict the frequency of metacognitive strategy use, and it is possible that learners perceive studying as something to be done in the near future to show results on a test, both in terms of perceived benefit [25] and cost. It is interesting to note that unlike cognitive strategies, the conditions of use appropriate to the situation discouraged the use of metacognitive strategies. Metacognition, as the term implies, is a higher-order cognition of the target cognition/behavior, and thus may be difficult for participants to evaluate. Therefore, metacognitive strategies are difficult to use, and the results of this study (i.e., CostSS predicting Used, may reflect confusion about judging the "appropriate use" of metacognitive strategies in a situation.

On a similar note, although the strengths of the correlations were smaller for metacognitive strategies than cognitive strategies, I'd be hesitant to compare these correlations because different students responded to the cognitive and metacognitive surveys, I believe. The manuscript currently suggests that "Thus, for the variance among multiple strategies within a learner, the factors causing the variance differed between cognitive and metacognitive strategies." (line 547). I'd rewrite this sentence for clarity and also temper this language because I don't think that you have evidence to directly conclude that the factors that influence cognitive and metacognitive strategy use are different.

 Thank you for your suggestions and remarks. I agree that the comment here could not be supported by the results of the paper. I have made the following correction.

BEFORE: page 16, line 547.

Thus, for the variance among multiple strategies within a learner, the factors causing the variance differed between cognitive and metacognitive strategies.

AFTER: page 16, line 568.

Thus, it was shown that cognitive and metacognitive strategies may differ in their conditional knowledge of costs that lead to variance in the degree of use. However, because of the different participants and their sample sizes, the results of Studies 1 and 2 cannot be directly compared, and the differences in the behavior of within-individual variance between cognitive and metacognitive strategies should be mentioned with data measured on the same sample of these strategies.

---

## [Editor Report · Decision Letter 2]

23 Jun 2022

PONE-D-21-02885R2A paradigm shift in learning strategy research: Illustration and example of a within-person examinationPLOS ONE

Dear Dr. Yamaguchi,

Thank you for submitting your manuscript to PLOS ONE. After careful consideration, we feel that it has merit but does not fully meet PLOS ONE’s publication criteria as it currently stands. Therefore, we invite you to submit a revised version of the manuscript that addresses the points raised during the review process.

Thank you for submitting your second revision of PONE-D-21-02885R2, entitled A paradigm shift in learning strategy research: Illustration and example of a within- person examination. To be transparent, I wanted to let you know that I was a reviewer of the previous versions of the manuscript and have taken over the Guest Editor role and will therefore be making a decision on the second revision of this manuscript. Both reviewers agreed that revision 1 was a substantial improvement from the original submission and recommended only minor additional revisions. Revision 2 incorporated these suggestions, which have further improved the clarity of the manuscript, particularly the interpretation of the results. Therefore, I have decided not to send revision 2 out for review again and only request minor revisions and I intend to accept the manuscript pending these revisions. These revisions are primarily for ease of reading and address language and terminology:

p. 19 line 709: Please use the term multivariate normality rather than multiple normalityTables 1 and 4: Please use an abbreviated strategy names/label such as “Understand Relationships” rather than the cs4 etc.I would recommend integrating the survey questions and responses into the text itself, rather than listing it at the end of the manuscript in the Supporting Information.The knowledge question asks participants to indicate if they were aware of the strategy. Throughout the manuscript, though, various terms such as “noticed” (p. 7 line 290) and “recognized” (p. 10 line 376) were used for this variable as well. These verbs don’t quite match the question. I would recommend use “knew of” or “were are of” instead.The cost questions specifically ask about whether using a strategy would be “troublesome” (pp. 19-20). “Cumbersome” (e.g., p. 14 line 402), “hard” (e.g., p. 16 line 563), and “difficult” (e.g., p. 16 line 536) were other words used to refer to costs. I know the survey questions were translated from Japanese for the manuscript, but I’m wondering if you could explain what the Japanese adjective for the cost questions would best translate to and what it would connote. Is it troublesome? Do cumbersome and hard work as synonyms as well? What I’m wondering is if the word that was used connotes anything about the strategy being logistically difficult (the way that the word cumbersome does) or is connoting something more along the lines of intellectually difficult (the way hard or difficult) might. In other words, were the cost questions likely having participants reflect on the practical effort of implementing a strategy (e.g., time, effort, resources) or was it more about the difficulty of the study experience itself (e.g., confusing, difficult, or tricky in the way that a challenging math problem can be). Clarifying the implied meaning of the Japanese adjective for that was used for “troublesome” in the cost survey questions could help readers understand which kinds of costs students generally avoid and could be relevant for designing future survey questions to tease apart different types of costs associated with a study strategy.Relatedly, the two suitable cost questions ask about using a strategy “only when it fits the situation.” However, there are places in the manuscript that mention whether the strategy is suitable for the content (e.g., abstract, p. 16 line 567). My understanding is that the actual survey questions didn’t ask about the suitability for the content, per se, but rather, the suitability for the situation in general? Is that accurate? Please clarify what the suitable questions were asking about. Also consider rephrasing “just study for now” (p. 11 line 425) accordingly.In general, I would be careful not to use quotation marks when paraphrasing the question or interpreting the results unless you are directly quoting the (translated) survey question (e.g., abstract, p. 14 line 502, p. 16 line 562).==============================

We look forward to receiving your revised manuscript.

Kind regards,

Hannah Hausman, Ph.D.

Guest Editor

PLOS ONE
---

## [Author Response · Author response to Decision Letter 2]

18 Jul 2022

Dr. Hannah Hausman

Guest Editor

PLOS ONE

Dear Dr. Hausman,

I would like to thank you for serving not only as a reviewer but also as a Guest Editor of my manuscript [PONE-D-21-02885, R1, and R2]. I am grateful for your advice to make my manuscript more readable. I have addressed each of your comments and suggestions as follows.

p. 19 line 709: Please use the term multivariate normality rather than multiple normality

RESPONSE: Thank you for pointing this out. I have corrected it.

BEFORE: p. 19 line 709

This study used MCMC methods to address multiple normality for estimating a large number of parameters, including variable effects with mixed effects models.

AFTER: p.21 line 748

The present studies used the MCMC method to address multivariate normality for estimating a large number of parameters, including variable effects with mixed-effects models.

Tables 1 and 4: Please use an abbreviated strategy names/label such as “Understand Relationships” rather than the cs4 etc.

RESPONSE: Thank you for your suggestion. I have labeled each of the abbreviated items. I have also associated this label with the strategy description in accordance with your next suggestion. In order to fit into the page layout, the table structure was varied: the means and standard deviations were combined in one cell, with the standard deviations in parentheses. The table title and note have also been partially changed accordingly. In the "Revised Manuscript with Track Changes" file, the old table was deleted because the serial numbers would shift if the old file was kept.

<Table1>

BEFORE: Label

cs1, cs2, cs3, cs4, cs5, cs6, cs7, cs8

AFTER: Label

Memorize Meaning, Understand Relationships, Memorize the Big Picture, Try to Understand, Ignore Origin, Ignore Meaning, Ignore the Big Picture, Just Write

BEFORE: Title

The mean (M) and standard deviation (SD) of each assessment item of cognitive strategies and the number of people with and without strategy knowledge in Study 1.

AFTER: Title

The mean and standard deviation (enclosed in parentheses) of each assessment item of cognitive strategies and the number of people with and without prior strategy knowledge in Study 1.

BEFORE: Note a

The cells in the "M" column show the number of people in "noticed (coded to 0.5)", and the cells in the "SD" column show the number of people in "did not notice (coded to -0.5)".

AFTER: Note a

The number at the top of the cell indicates the number of respondents who indicated they knew about the strategy (coded as 0.5), while the number at the bottom of the cell indicates the number of respondents who indicated they did not know about the strategy (coded as -0.5).

<Table 4>

BEFORE: Table 4

ms1. ms2, ms3, ms4, ms5, ms6, ms7

AFTER: Table 4

Predict Accomplishment, Clarify Understanding, Grasp Understanding, Evaluate Understanding, Study Planning, Appropriate Studying, Set Achievement

BEFORE: Title

The mean (M) and standard deviation (SD) of each assessment item of metacognitive strategies and the number of people with and without strategy knowledge in Study 2.

AFTER: Title

The mean and standard deviation (enclosed in parentheses) of each assessment item of metacognitive strategies and the number of people with and without prior strategy knowledge in Study 2.

BEFORE: Note a

The cells in the "M" column show the number of people in "noticed (coded to 0.5)", and the cells in the "SD" column show the number of people in "did not notice (coded to -0.5)".

AFTER: Note a

The number at the top of the cell denotes the number of respondents who answered indicated they knew about the strategy (coded as 0.5), while the number at the bottom of the cell indicates the number of respondents who indicated they did not know about the strategy (coded as -0.5).

I would recommend integrating the survey questions and responses into the text itself, rather than listing it at the end of the manuscript in the Supporting Information.

RESPONSE: Thank you for your suggestion. I have removed the "Supporting Information" segment and inserted those sentences into the "Measures" sections of Studies 1 and 2.

BEFORE: Supporting Information

AFTER: deleted

BEFORE: N/A

AFTER: p. 7 line 281

The items on cognitive strategy in Study 1 are as follows (as the survey instructions and questionnaire items are in Japanese [31] http://doi.org/10.15002/00010874, the author asked an English editing service editage) to perform back-translation to check the accuracy of the translation):

[Eight items]

The order in which the items are presented here corresponds to Table 1. Numbers in parentheses after the description indicate the items’ order of presentation in the questionnaire, whereas the labels in square brackets match the headings in Table 1. Questionnaires were randomly distributed in reverse order due to counterbalancing.

BEFORE: N/A

AFTER: p. 8 line 305

[New Paragraph]

Participants were required to respond to seven items for each strategy. The teaching and response items presented to the participants were as follows:

The following items indicate ways of studying that you may or may not use in your learning process. For each item, please circle a number for the following six: {1} How well does this apply to your current way of studying? {2} How cumbersome do you think it would be to use this method at all times, regardless of the circumstances, in order to improve your score on your next test? {3} How cumbersome do you think it would be to use this method only when it suits the situation in order to score points on your next test? {4} How cumbersome do you think it would be to use this method at all times, regardless of the circumstances, in continuing your future learning? {5} How cumbersome do you think it would be to use this method only when it suits the situation in continuing your future learning? {6} How effective do you think this method is? After “How well does this apply to your current way of studying?” for a given method, you will be asked {7} whether you “(A) were aware of this strategy beforehand” or “(B) were not aware about this strategy until reading about it on this questionnaire.” Please circle the corresponding letter. There are no right or wrong answers, so please answer based on what you think is accurate.

[Seven values]

BEFORE: N/A

AFTER: p. 15 line 516

The items of the metacognitive strategy in Study 2 are as follows:

[Seven items]

The order in which the items are presented here corresponds to Table 4. Numbers in parentheses after the description indicate the order of presentation in the questionnaire, whereas the labels in square brackets match the headings in Table 4. Questionnaires were also randomly distributed in reverse order due to counterbalancing.

The knowledge question asks participants to indicate if they were aware of the strategy. Throughout the manuscript, though, various terms such as “noticed” (p. 7 line 290) and “recognized” (p. 10 line 376) were used for this variable as well. These verbs don’t quite match the question. I would recommend use “knew of” or “were are of” instead.

RESPONSE: Thank you for pointing this out. I have modified these lines to improve their clarity.

BEFORE: p. 7 line 290

Knowledge regarding strategy was binary (“noticed” coded 0.5, or “did not notice” coded -0.5), and the other aspects were scored on a Likert scale from 1 (“Not at all”) to 6 (“Very true of me”).

AFTER: p. 9 line 326

Knowledge regarding strategy was binary (“knew of,” coded as 0.5, or “did not know of,” coded as -0.5), whereas the other aspects were scored on a Likert scale ranging from 1 (“Not true at all”) to 6 (“Very true”).

BEFORE: p. 10 line 376

First, knowledge regarding strategies (Knowledgeij)was coded “0.5” (for recognized) or “-0.5” (for did not recognize).

AFTER: p. 11 line 411

First, knowledge regarding strategies (Knowledgeij)was coded as 0.5 (knew about the strategy) or -0.5 (did not know about the strategy).

The cost questions specifically ask about whether using a strategy would be “troublesome” (pp. 19-20). “Cumbersome” (e.g., p. 14 line 402), “hard” (e.g., p. 16 line 563), and “difficult” (e.g., p. 16 line 536) were other words used to refer to costs. I know the survey questions were translated from Japanese for the manuscript, but I’m wondering if you could explain what the Japanese adjective for the cost questions would best translate to and what it would connote. Is it troublesome? Do cumbersome and hard work as synonyms as well? What I’m wondering is if the word that was used connotes anything about the strategy being logistically difficult (the way that the word cumbersome does) or is connoting something more along the lines of intellectually difficult (the way hard or difficult) might. In other words, were the cost questions likely having participants reflect on the practical effort of implementing a strategy (e.g., time, effort, resources) or was it more about the difficulty of the study experience itself (e.g., confusing, difficult, or tricky in the way that a challenging math problem can be). Clarifying the implied meaning of the Japanese adjective for that was used for “troublesome” in the cost survey questions could help readers understand which kinds of costs students generally avoid and could be relevant for designing future survey questions to tease apart different types of costs associated with a study strategy.

RESPONSE: Thank you for pointing out the lack of clarity in the wording. To quote your expression, I believe that "the practical effort of implementing a strategy (e.g., time, effort, resources)" is appropriate for my intended perceived cost of a learning strategy. The appropriate adjective placement for such costs was "intellectually difficult" for me, but the appropriate expression for this paper seems to be "cumbersome".

BEFORE: troublesome, hard, difficult

AFTER: cumbersome

Relatedly, the two suitable cost questions ask about using a strategy “only when it fits the situation.” However, there are places in the manuscript that mention whether the strategy is suitable for the content (e.g., abstract, p. 16 line 567). My understanding is that the actual survey questions didn’t ask about the suitability for the content, per se, but rather, the suitability for the situation in general? Is that accurate? Please clarify what the suitable questions were asking about. Also consider rephrasing “just study for now” (p. 11 line 425) accordingly.

RESPONSE: Thank you for your observation. I agree with your interpretation. This paper’s research design could only refer to the subjective situation of the learner.

BEFORE: Abstract

The analysis revealed that in terms of cognitive strategies (Study 1), learners avoided using learning strategies that they perceived as high cost to "use for the next test" and "keep using the strategy anyway, regardless of the content". On the other hand, in metacognitive strategies (Study 2), students avoided using learning strategies that they perceived as costly to "use for the next test" and "use as appropriate for the content".

AFTER: Abstract

The analysis revealed that, in terms of cognitive strategies (Study 1), learners avoided using learning strategies perceived to have a high cost in the short term; however, they used cognitive strategies regardless of the circumstances. Furthermore, regarding metacognitive strategies (Study 2), students avoided using learning strategies that they perceived as costly to use in the short term and used them only when they suited the circumstances.

BEFORE: p. 16 line 567

For cognitive strategies, the higher the subjective value of "it's hard to keep using it anyway for the next test," the less likely they were to use the strategy; On the other hand, metacognitive strategies showed the same correlation as cognitive strategies in short-term "when" conditional knowledge, such as "use it for the next test," but the higher the cost of "how" conditional knowledge, such as "use it according to the content," the less likely they were to use the strategy.

AFTER: p. 18 line 612

For cognitive strategies, the higher the subjective cost of a given strategy, the less likely participants were to use it; 

Conversely, while metacognitive strategies showed the same correlation as cognitive strategies vis-à-vis temporal conditional knowledge (i.e., using a given strategy in the short term, such as in a future test), the higher the cost of a strategy’s methodology-related conditional knowledge (using a given strategy if the circumstances are suitable), the less likely were learners to use it.

BEFORE: p. 11 line 425

Therefore, a short time condition (e.g., the next test) and a condition in which participants did consider the appropriate use of the cognitive strategy (e.g., just study for now) may have predicted the reduced use of the cognitive strategy.

AFTER: p. 13 line 460

Therefore, a short timeframe (e.g., the next test) and a condition in which participants did not have to consider the appropriate use of the cognitive strategy (i.e., they did not take into account the circumstances) may have predicted the reduced use of the cognitive strategy.

In general, I would be careful not to use quotation marks when paraphrasing the question or interpreting the results unless you are directly quoting the (translated) survey question (e.g., abstract, p. 14 line 502, p. 16 line 562).

RESPONSE: Thank you for pointing this out. I have corrected them. I apologize for the inconvenience.

BEFORE: Abstract

The analysis revealed that in terms of cognitive strategies (Study 1), learners avoided using learning strategies that they perceived as high cost to "use for the next test" and "keep using the strategy anyway, regardless of the content". On the other hand, in metacognitive strategies (Study 2), students avoided using learning strategies that they perceived as costly to "use for the next test" and "use as appropriate for the content".

AFTER: Abstract

The analysis revealed that, in terms of cognitive strategies (Study 1), learners avoided using learning strategies perceived to have a high cost in the short term; however, they used cognitive strategies regardless of the circumstances. Furthermore, regarding metacognitive strategies (Study 2), students avoided using learning strategies that they perceived as costly to use in the short term and used them only when they suited the circumstances.

BEFORE: p. 14 line 502.

First, for the effect of the four perceived costs, CostSS, “is cumbersome to use properly for the next exam,” suppressed learning strategy use (gamma20 = -0.208 [-0.365–-0.038]). 

AFTER: p. 15 line 551

First, for the effect of the four perceived costs, CostSS, “[this strategy] is cumbersome to use properly for the next exam,” suppressed learning strategy use (gamma20 = -0.208 [-0.365–-0.038]). 

BEFORE: p. 16 line 562

For example, between-person correlations are interpreted as "the more troublesome the strategy is, the less likely it is to be used," while within-person correlations are interpreted as "the more troublesome a strategy is, the less likely it is to be used compared to other strategies that are not troublesome."

AFTER: p. 18 line 630

For example, between-person correlations are interpreted as "the more cumbersome the strategy is, the less likely it is to be used," while within-person correlations are interpreted as "the more cumbersome a strategy is, the less likely it is to be used, compared with less cumbersome strategies."

---

## [Editor Report · Decision Letter 3]

31 Aug 2022

A paradigm shift in learning strategy research: Illustration and example of a within-person examination

PONE-D-21-02885R3

Dear Dr. Yamaguchi,

We’re pleased to inform you that your manuscript has been judged scientifically suitable for publication and will be formally accepted for publication once it meets all outstanding technical requirements.

Kind regards,

Hannah Hausman, Ph.D.

Guest Editor

PLOS ONE

Additional Editor Comments (optional):

I have received your revisions and am pleased to accept your manuscript for publication. I want to acknowledge the hard work that you put into editing this paper and I am confident that PLOS ONE readers will be eager to learn from your new approach to assessing when and why students implement various study strategies.
---

## [Editor Report · Acceptance letter]

5 Sep 2022

PONE-D-21-02885R3 

A paradigm shift in learning strategy research: Illustration and example of a within-person examination 

Dear Dr. Yamaguchi:

I'm pleased to inform you that your manuscript has been deemed suitable for publication in PLOS ONE. Congratulations! Your manuscript is now with our production department. 

Kind regards, 

on behalf of

Dr. Hannah Hausman 

Guest Editor

PLOS ONE